# Radiative and chemical implications of the size and composition of aerosol particles in the existing or modified global stratosphere

Daniel M. Murphy[1], Karl D. Froyd[1,2], Ilann Bourgeois[1,2], Charles A. Brock[1], Agnieszka Kupc[1,2,3], Jeff Peischl[l,2], Gregory P. Schill[1,2], Chelsea R. Thompson[1,2], Christina J. Williamson[1,2], Pengfei Yu[4]

[1]NOAA Chemical Sciences Laboratory, Boulder, CO 80305, USA
[2]Cooperative Institute for Research in Environmental Sciences, University of Colorado, Boulder, CO 80309, USA
[3]Faculty of Physics, Aerosol Physics and Environmental Physics, University of Vienna, 1090 Vienna, Austria
[4]Institute for Environment and Climate Research, Jinan University, Guangzhou, China

*Correspondence to*: Daniel Murphy (daniel.m.murphy@noaa.gov)

**Abstract.** The size of aerosol particles has fundamental effects on their chemistry and radiative effects. We explore those effects using aerosol size and composition data in the lowermost stratosphere along with calculations of light scattering. In the size range between about 0.1 and 1.0 µm diameter (accumulation mode), there are at least two modes of particles in the lowermost stratosphere. The larger mode consists mostly of particles produced in the stratosphere and the smaller mode consists mostly of particles transported from the troposphere. The stratospheric mode is similar in the Northern and Southern hemispheres whereas the tropospheric mode is much more abundant in the Northern Hemisphere. The purity of sulfuric acid particles in the stratospheric mode shows that there is limited production of secondary organic aerosol in the stratosphere, especially in the Southern Hemisphere. Out of eight sets of flights sampling the lowermost stratosphere (four seasons and two hemispheres) there were three with large injections of specific materials: volcanic, biomass burning, or dust. The stratospheric and tropospheric modes have very different roles for radiative effects on climate and for heterogeneous chemistry. Because the larger particles are more efficient at scattering light, most of the radiative effect in the lowermost stratosphere is due to stratospheric particles. In contrast, the tropospheric particles can have more surface area, at least in the Northern Hemisphere. The surface area of tropospheric particles could have significant implications for heterogeneous chemistry because these particles, which are partially neutralized and contain organics, do not correspond to the substances used for laboratory studies of stratospheric heterogeneous chemistry. We then extend the analysis of size-dependent properties to particles injected into the stratosphere, either intentionally or from volcanoes. There is no single size that will simultaneously maximize the climate impact relative to the injected mass, infrared heating, potential for heterogeneous chemistry, and undesired changes in direct sunlight. In addition, light absorption in the far ultraviolet is identified as an issue requiring more study for both the existing and potentially modified stratosphere.

## 1 Introduction

Stratospheric particles have been studied for over 60 years (Junge and Manson, 1961; reviewed by Kremser et al., 2016). The stratospheric aerosol layer has a maximum in mixing ratio between about 20 and 25 km altitude. However, the larger air density

at lower altitudes means that the majority of the mass of stratospheric aerosol is in the lowermost stratosphere, below the maximum in mixing ratio (Yu et al., 2016).

Various trends have been reported for the background stratospheric aerosol at times not influenced by major volcanic eruptions. Deshler et al. (2006) concluded there was little long-term change in background stratospheric aerosol from 1971 to 2004. Hofmann et al. (2009) found an increasing trend after 2000, and Friberg et al. (2014) found an increasing trend from 1999 to 2008. There has been recognition that moderate volcanic eruptions frequently influence the stratospheric aerosol and true

nonvolcanic "background" concentrations are not necessarily present just because there has been no Pinatubo-scale eruption (Solomon et al., 1996, Vernier et al. 2011). Different altitudes may exhibit different trends (Khaykin et al., 2017). Moderate volcanic eruptions tend to mask trends in the non-volcanic background (Kremser et al., 2016). The data shown here reinforce the notion of modest but frequent perturbations to the lower stratosphere.

The overall circulation of air in the stratosphere, with rising air in the tropics and descending air in the extratropics, is mostly fed by air entering at the tropical tropopause. The lowermost stratosphere is a region at middle and high latitudes between the local tropopause and slightly above the altitude of the tropical tropopause. Air in this lowermost stratosphere is affected by both downward motion in the extratropical stratosphere and adiabatic mixing with the troposphere (Holton et al., 1995). The tropospheric influence can extend as high as about 450 K potential temperature (Rosenlof et al., 1997). All of the data described

here are in the lowermost stratosphere and show the influence of both air from higher in the stratosphere and air from the troposphere.

The chemical composition of particles in the lower stratosphere has been measured by several techniques. Impactor samples collected from the CARIBIC platform have been quantitatively analyzed for N, O, S, K, Fe, and other elements (Nguyen and

Martinsson, 2007; Friberg et al, 2014; Martinsson et al., 2019). The moles of oxygen were approximately four times the moles of sulfur, plus about 0.2 times the moles of carbon, indicating $SO_4$ in sulfate and sulfuric acid plus some contribution from oxygenated organics. Much of the detailed information on the chemical composition of aerosols in the lower stratosphere has come from the Particle Analysis by Laser Mass Spectrometry (PALMS) instrument (Murphy et al., 1998), which analyses individual particles. These data show that most particles larger than about 110 nm fall into three distinct types: sulfuric acid

with or without meteoric metals and internally mixed organic-sulfate particles from the troposphere (Murphy et al., 2014). Because the pure sulfuric acid particles do not contain biomass burning residues, they are not simply tropospheric particles that have lost organics after entering the stratosphere (Murphy et al., 2007). Recent data from another single particle mass spectrometer have found comparable abundances of sulfuric acid particles with meteoric metals (Schneider et al., in discussion).


We extend the previous results to show that the mixed organic-sulfate particles from the troposphere are generally smaller than both types of sulfuric acid particles. We then show how this size difference has significant implications for light scattering and heterogeneous chemistry.

## 2 Methods

This paper includes data in the lowermost stratosphere from the Atmospheric Tomography (ATom) mission with deployments in four seasons during 2016 to 2018. Although the ATom mission was not specifically designed to sample the stratosphere, it encountered stratospheric air in both the Northern and Southern Hemispheres during its regular vertical profiles. Stratospheric air was encountered periodically at altitudes greater than about 7 km and latitudes poleward of about 30 degrees north or 35 degrees south (Figure S1 and Table S1). Because the NASA DC8 aircraft has a ceiling of about 12 km, stratospheric air was always associated with low tropopauses, sometimes in tropopause folds. We therefore use ozone rather than altitude as the primary definition of how far into the stratosphere measurements were taken. If there were mixing with a great deal of tropospheric air it would reduce the ozone concentration below the thresholds we set for stratospheric measurements. Data below 7 km altitude were excluded just in case there were unusually high ozone concentrations at low altitude.

Size distributions for accumulation-mode particles were measured using two modified commercial laser optical particle spectrometers, an ultra-high sensitivity aerosol spectrometer (UHSAS; Droplet Measurement Technologies, Longmont, USA) from 0.07 to 0.6 μm diameter and a laser aerosol spectrometer (LAS, TSI Inc., St. Paul, USA) from 0.6 to ~4.8 μm diameter (Kupc et al., 2018; Brock et al., 2019). The diameters are based on calibration by ammonium sulfate particles. The size resolution of the reported data is 20 bins per decade of particle size. Data are recorded at 1s intervals although averaging is needed in the stratosphere to improve counting statistics for particles in the LAS size range (Brock et al., 2019). Ozone measurements are described by Bourgeois et al., 2020.

Particle composition was measured with the Particle Analysis by Laser Mass Spectrometry (PALMS) instrument (Thomson et al., 2000; Froyd et al. 2019). A pressure-controlled aerodynamic focusing inlet brings particles into a vacuum where they cross two continuous laser beams. The transit time between the beams measures the aerodynamic diameter of each particle. The aerodynamic diameters are under vacuum conditions with most particles much smaller than the mean free path at the inlet exhaust. Transition flow corrections are considered. Transit times were calibrated to known particle sizes before and after every field deployment. A 193 nm pulse from an excimer laser is triggered when a particle arrives at the second laser beam. Either positive or negative ions are analyzed with a time-of-flight mass spectrometer. The polarity was switched every few minutes. Most of the data shown here are from positive ion spectra. Negative ion spectra do not distinguish sulfuric acid with and without meteoric metals because the metal ions only appear in the positive ion spectra.

The optical size distributions are combined with the PALMS single-particle composition data for particles larger than 100 nm to create size distributions that are resolved by composition. In each size range, the number of particles is obtained from the optical size distributions and the fraction of particles with different compositions is obtained from PALMS. This combination requires converting the PALMS aerodynamic diameters to correspond to the optical diameters (Froyd et al., 2019). The composition-resolved size distributions presented here use wider size bins than the native optical particle counter resolution

but narrower size bins than the standard ATom products (Froyd et al., 2019). The narrower bins are possible because of improved statistics after averaging over all of the data within a specific band of ozone and latitude, even if those data were not contiguous in time.

For the purpose of this study particles are classified into four basic categories: sulfuric acid with and without meteoric metals,

mixed organic-sulfate particles, and other particles including dust. Example spectra are shown in Murphy et al. (2014). When appropriate further distinctions can be made, such as separating biomass burning particles from other mixed organic-sulfate particles. The sulfuric acid particles, either with or without meteoric metals, originate in the stratosphere. The organic-sulfate and most other particles such as dust originate in the troposphere or at the surface and mix into the stratosphere.

Sources of uncertainty include particle statistics, identifying particle types, the particle volume from the optical particle counters, and combining the PALMS spectra with the optical size distributions Overall statistics are excellent, with approximately 800,000 single particle mass spectra acquired above 7 km altitude during ATom. Of those, approximately 78,000 were positive ion spectra at ozone concentrations larger than 250 ppbv (Table S1). However, low ambient number densities for large particles result in fewer mass spectra of particles larger than about 1 μm, leading to statistical noise for large

particles visible in the figures in this paper. Except for ATom4 Northern Hemisphere, particles larger than 1 μm contributed only a very small fraction of the aerosol volume in the lower stratosphere. Classification uncertainty depends on the type of particle. The meteoric-sulfuric particles are very distinctive. From manually examining mass spectra we estimate that only a few percent are incorrectly classified. Sulfuric acid particles are more difficult to classify because some tropospheric particles with low organic content are similar to stratospheric sulfuric acid particles. The criteria used here probably err on the side of

underestimating tropospheric particles in the lower stratosphere, especially in the Southern Hemisphere. Using various criteria for separating organic-sulfate and sulfuric acid particles suggests a classification uncertainty of up to 25% for those particle types in the lower stratosphere. With sufficient averaging (minutes), the volume derived from optical size distributions has an uncertainty propagated from size and flow uncertainties of about +13/-28% in the accumulation mode and up to +/- 50% above 1 μm (Kupc et al., 2018; Brock et al., 2019). Excellent agreement between extinctions calculated from the size distributions

and independent extinction measurements indicates that systematic errors may actually be less than this (Brock et al., 2019). Because PALMS measures aerodynamic diameters, mapping the PALMS spectra onto the optical size distributions requires particle density and shape (Froyd et al., 2019). These are well known for sulfuric acid particles, less so for some particles such as dust. How uncertainty in density affects the combination of PALMS and optical data depends on the shape of the size

distribution and the number of types of particles. In cases with flat size distributions or where only one dominant type of
particle is present, uncertainty in density introduces very little additional uncertainty when combing PALMS data with size distributions. The uncertainty is larger if there are several types of particles present in a size range with rapidly changing particle volume versus size. Statistical uncertainty in the fractional organic content of mixed particles is less than 10% if at least dozens of mass spectra are averaged (Froyd et al., 2019).

Calculations of light scattered back to outer space are made using Mie scattering for an optically thin layer uniformly spread over a sunlit hemisphere as described by Murphy (2009), except that these calculations use an atmospheric transmittance appropriate for approximately 11 km altitude (Arvesen, 1969). Changing the solar spectrum over the entire range from top-of-atmosphere to surface gives qualitatively similar results. At the low relative humidities in the stratosphere, water uptake is less important for optical properties than it is in the troposphere. The mean relative humidity for the ATom data at ozone > 250
ppbv was less than 10%.

A sectional aerosol model (CARMA) coupled with the NSF/DOE Community Earth System Model (CESM) is used in the study to simulate the composition and size distributions of stratospheric aerosols (Yu et al., 2015; Toon et al., 1988). CESM-CARMA tracks two external-mixed groups of aerosols. The first group consists of pure sulfate particles (formed through
nucleation and condensation of water vapor and sulfuric acid) with 20 size bins ranging from 0.4 nm to 2.6 µm in diameter; the second group consists of internal mixed aerosols (containing condensed sulfate, organics, black carbon, salt and dust) with 20 size bins from 0.1 µm to 17 µm. The model does not explicitly separate meteoric-sulfuric particles from other sulfuric acid particles. The model includes secondary organic chemistry (Yu et al., 2015). The model is run at a horizontal resolution of 1.9° (latitude) x 2.5° (longitude). It has 56 vertical layers from the surface to 1.8 hPa with a vertical resolution of ~1 km near
the tropopause.

**3 Composition-resolved size distributions in the stratosphere**

Figure 1 shows the composition-resolved size distributions measured in the lowermost stratosphere for the four ATom
deployments, separated by the Northern and Southern Hemisphere. The data are for ozone between 250 and 400 ppbv. This range of ozone is chosen to be definitely in stratospheric air and to include data from both hemispheres on all four deployments. For comparison, the median ozone concentration at the lapse rate thermal tropopause was slightly less than 100 ppbv. On each panel the thick black line is the size distribution from the optical particle counters. At each size the fraction of particle types from PALMS is shaded. A number of features in Figure 1 are worthy of comment.
The volume distributions show a peak near 400 nm diameter and another peak, or at least a shoulder, near 180 or 200 nm. These sizes are both within what is considered the accumulation mode. The composition reveals why there are two modes.

Indeed, without the composition it would be difficult to be sure that there were two separate modes. For example, the Wyoming particle counters used on stratospheric balloon flights (Deshler et al., 2003) do not clearly resolve the modes.


The 400-nm mode is from sulfuric acid particles produced in the stratosphere, especially those with meteoric metals. The size of these meteoric-sulfuric particles is extremely consistent through both hemisphere and the four deployments (Table 1). The primary source of sulfuric acid in the stratosphere, oxidation of carbonyl sulfide, is similar in the two hemispheres. The meteoric-sulfuric particles also have a narrow size distribution, with a typical geometric standard deviation of about 1.4 when fit with a log-normal distribution. This is consistent with condensational growth, which tends to lead to narrow size distributions because smaller particles have relatively more surface area for condensation. Sulfuric acid particles without meteoric material have more diverse sizes except for the volcanically influenced ATom1 Southern Hemisphere, when the sulfuric acid particles had a narrow size distribution very similar to the meteoric-sulfuric particles.

The smaller mode near 200 nm is from mixed organic-sulfate particles that have mixed into the stratosphere. The mass spectra of particles in the smaller mode are essentially identical to those of particles in the upper troposphere (Figure S2). Unlike studies that relied solely on bulk composition (e.g. Martinsson et al., 2019), we identify the tropospheric contribution based on the mass spectra of individual particles (Murphy et al., 2007). Comparing the Northern and Southern Hemispheres in Figure 1, the concentration of the smaller mode is larger in the Northern Hemisphere. The upper troposphere in the Southern Hemisphere has generally lower aerosol concentrations, so mixing in a given amount of tropospheric air will bring in fewer particles than the same amount of mixing in the Northern Hemisphere.

Of the eight cases in Figure 1, three have much higher aerosol concentrations than the others, for three very different reasons. The ATom1 wintertime Southern Hemisphere had a mode of pure sulfuric acid particles. These were most likely produced from $SO_2$ injected into the stratosphere by the Calbuco eruption in April 2015, about 16 months before the measurements (Bègue et al., 2017). The sulfuric acid particles are remarkably pure, except for associated water. This is consistent with previous data on aerosol volatility and infrared spectra after the eruption of Mt. Pinatubo (Deshler et al., 1992; Grainger et al., 1993). The size distribution is also quite narrow, with a geometric standard deviation $\sigma_g$ of 1.29 for just the sulfuric acid particles (Table 1). Satellite aerosol retrievals may use wider size distributions. For example, Bauman et al. (2003) use a lookup range for $\sigma_g$ of 1.3 to 2.3 and show that retrievals may not find solutions if $\sigma_g \approx 1.1$. The size distribution of these volcanic particles suggests caution in satellite and lidar retrievals of stratospheric aerosols dominated by one source of particles grown by condensation of sulfuric acid. As can be seen in Figure 1, overall size distributions typically are not this narrow because they are broadened with particles from multiple sources.

The ATom3 Northern Hemisphere had a large tropospheric organic-sulfate contribution. More detailed composition shows that these included a large fraction of biomass burning particles. The ATom3 flights were about two months after Canadian

fires produced a massive injection of smoke into the stratosphere, with additional injection rising through the tropopause due to diabatic heating (Torres et al., 2020). A separate paper is in preparation about the in-situ data from this wildfire event. The ATom4 springtime Northern Hemisphere had both a large contribution from organic-sulfate particles and a remarkable amount

of dust at and above the tropopause. Concentrations at the tropopause often reached several micrograms per standard cubic meter. The dust was very widespread: it was measured over both the north Atlantic and Pacific Oceans over more than 40 degrees of latitude. This may be Asian dust and pollution carried to high altitude in an event similar to that described by Huang et al. (2008). The ATom4 and Huang et al. events were both in May. A separate paper is also planned about this dust event.

Figure 2 shows the CESM/CARMA model results for the ATom2 flights for the same 250 – 400 ppbv range of ozone as the data in Figure 1. ATom2 is chosen because neither hemisphere was perturbed by volcanic sulfate, biomass burning, or dust. The model reproduces the tropospheric mode well in the Northern Hemisphere but overestimates it in the Southern Hemisphere. The model reproduces the total volume of stratospheric particles well in both hemispheres but the modeled diameter of these particles is too small. A possible reason is that the model does not include meteoric smoke particles on which

sulfuric acid can condense. That is, the model treats both the meteoric-sulfuric and sulfuric acid particles observed by PALMS as a single type. Figure S3 compares the Northern Hemisphere ATom data to Wilson et al. (2008). Consistent with the ATom observations, the tropospheric particle mode at about 200 nm is in some cases distinct from the larger stratospheric mode.

      Figure 3 shows composition-resolved size distributions further into the stratosphere with ozone between 500 and 850 ppbv.
Only the Northern Hemisphere during ATom2 and ATom4 had significant amounts of PALMS data in this ozone range. For ATom2, the primary difference at higher ozone was more meteoric-sulfuric particles, a result also found by Schneider et al. (in discussion). The large mixing event of dust and other tropospheric particles during ATom2 barely affected "altitudes" of more than 500 ppbv ozone. Figure 4 has some of the data in Figures 1 and 3 replotted to emphasize the size distributions of each component. The size distributions of the meteoric-sulfuric particles are extremely consistent. The size distributions of the

sulfuric acid particles without meteoric content are usually broader, possibly indicating more diverse sources. One exception is the very narrow size distribution of the particles after the Calbuco eruption. The ATom3 Northern Hemisphere has a mode of larger organic particles from the pyrocumulus injection event. The dust event in the Northern Hemisphere during ATom4 also brought up organic-sulfate and sulfuric particles, or their precursors, from near the surface as well as dust.

Figure 4 can also be used to illustrate what features of the size distributions are more certain or uncertain. The positions and widths of the modes are robust, as are broad features such as the presence of a bimodal organic-sulfate distribution for ATom3. Narrow variations in the distributions, such as the minimum near 0.55 μm in the ATom3 distribution, could be due to gain stitching or Mie scattering effects in the optical particle counters. In the bottom panel, the increased concentration of particles larger than 1 μm during ATom4 compared to the other deployments is robust but the sharpness of the increase at 2 μm could

easily be due to limited statistics for large particles.

## 4 Vertical profiles

The shaded regions in Figure 1 can be integrated over all sizes to determine the volume associated with each type of particle, then multiplied by a density to determine the mass. Figure 5 shows vertical profiles of the volume concentrations for the meteoric sulfuric particles and organic-sulfate particles. As expected for a high-altitude source, the concentration of meteoric-sulfuric particles increases with ozone concentration. Except for the ATom3 wildfire biomass burning injection, the concentration of organic-sulfate particles decreased with increasing ozone or, in one case, stayed roughly constant. The concentration of the meteoric-sulfuric particles is fairly consistent between hemispheres and deployments. More measurements are needed to see if the small seasonal differences are persistent in other years. In contrast, the concentration of organic-sulfate particles was larger in the Northern Hemisphere than in the Southern Hemisphere and varied considerably between deployments. It is worth noting that the highest concentration of tropospheric particles in each hemisphere, ATom3 for the Southern Hemisphere and ATom4 for the Northern Hemisphere, were both observed during local springtime.

Figure 6 shows the ratio of the $C^+$ peak, an indicator of organic content, to two peaks indicative of sulfate or sulfuric acid. The top axis gives an approximate mass fraction of organics adapted from calibrations described by Froyd et al. (2019). The vertical axis of ozone serves as a measure of distance into the stratosphere. The organic content is separated by particle type, something not possible with bulk analysis. That the stratospheric and tropospheric particle compositions remain distinct implies that there is very limited redistribution of semi-volatile organics between particles. Like most upper tropospheric particles, the organic-sulfate particles are internally mixed with on average about 40 to 80 percent organic material by mass. There is little variation with ozone, indicating a long lifetime for the organic material as well as little uptake of sulfuric acid. The latter is consistent with most of the sulfuric acid coming from carbonyl sulfide above 20 km rather than $SO_2$ near the tropopause (Kremser et al., 2016; Rollins et al., 2017). That there are two types of particles with different compositions present in the same air also means that any semi-volatile gas-phase organics and ammonia cannot be in equilibrium with both types of particles. Gas-phase ammonia concentrations must be extremely low or else the stratospheric sulfuric acid particles would be slowly neutralized. Even small amounts of gas-phase ammonia can strongly modulate new particle formation in the lower stratosphere (Williamson et al., 2021).

There is some limited uptake of organics onto the stratospheric particles, although the maximum organic concentration is still much less than for tropospheric particles. Meteoric-sulfuric particles definitely formed in the stratosphere, so any significant organic content indicates net uptake of organics. Their organic content grows as the particles descend to the lowermost stratosphere and the upper troposphere. The meteoric-sulfuric particles contain much less than 1% organic mass at altitudes with ozone greater than 500 ppbv and as much as 2 to 4% near the tropopause in the Northern Hemisphere. Such limited

formation of secondary organic mass in the lowermost stratosphere is consistent with previous PALMS measurements (Murphy et al., 2007).

A new finding from ATom is that there is a very distinct and consistent difference between the hemispheres in the small amount of organic content that does form in the meteoric-sulfuric acid particles. Since the particles start from similar formation processes much higher in the stratosphere, we conclude that there is more condensable or reactive organic vapor in the Northern Hemisphere lower stratosphere. This could be either gas phase species mixed from the troposphere or semi-volatile organics transferring from organic-sulfate particles.

Adding a few percent mass to the meteoric-sulfuric particles represents a very small amount of organic vapor. Without knowing uptake coefficients the amount of vapor cannot be determined uniquely, but a representative calculation is that one or two pptv of an organic gas-phase species with molecular weight of about 100 daltons that reacts with sulfuric acid on every collision would add few percent mass to a 450 nm particle in a few months. The same order of magnitude can be obtained by noting from Figure 5 that at 200 ppbv ozone there is about 100 ng standard $m^{-3}$ of meteoric sulfuric particles. 1% by mass of these particles corresponds to about a part per trillion by mass of air. We conclude that an order of magnitude for highly condensable organic vapor in the lowermost stratosphere is a few parts per trillion in the Northern Hemisphere and less in the Southern Hemisphere. A less reactive or condensable organic molecule could be present at a correspondingly higher concentration.

**5 Radiative and chemical implications**

The different sizes of the sulfuric acid and organic-sulfate particles lead to substantial differences in their radiative and chemical effects. Important properties are the amount of infrared heating, the amount of light scattered, implications for photolysis, the surface area available for heterogeneous chemistry, and the mass sedimentation rate.

A key part of the radiative implications is the efficiency of light scattering as a function of particle size. Figure 7 shows the mass scattering efficiency as a function of particle size averaged over the solar spectrum and a sunlit Earth. Calculations are for a real refractive index of 1.45 and minimal absorption. Atmospheric extinction is determined by the solid total scattering curve. Much of the light scattered by particles continues downward to the Earth and so does not directly affect climate. Separating out the light scattered to outer space (dashed curve) gives a maximum that is slightly broader and shifted to smaller sizes than light extinction. Over much of the size range of particles that scatter light efficiently, only about 1/5 of the light that is scattered goes to outer space; the remainder becomes diffuse light. This is a reason for the large increases in diffuse light (with decreases in direct sunlight) after volcanic eruptions (Murphy, 2009).

Figure 7 also shows an estimate of the importance of infrared absorption. Infrared absorption is estimated by adapting a calculation from Lacis et al. (1992), who showed that heating exceeded shortwave cooling for several different types of particles when their diameters were more than about 4 µm. Since infrared absorption per unit mass is almost independent of size, an approximate net cooling was estimated by subtracting a constant from the calculated shortwave scattering per unit mass to space such that the net was zero at 4 µm diameter. The vertical scale of the net cooling curve is consistent with the scattering curves. For example, at peak scattering efficiency near 0.5 µm, infrared effects reduce the cooling of the Earth by stratospheric aerosol by 10 to 15%.

The top panel of Figure 8 replots the chemically resolved size distribution for ATom2 in the wintertime Northern Hemisphere. This was chosen as an example because it is similar to several other locations and seasons such as ATom1 summertime Northern Hemisphere and ATom3 springtime Southern Hemisphere. There was more sampling time in the ATom2 Northern Hemisphere stratosphere so the particle statistics are better, with mass spectra of about 10,000 particles. In Figure 8 the two sulfuric acid particle categories have been combined. Of the particles larger than 0.1 µm diameter, about 39% of the volume was organic-sulfate particles from the troposphere and 61% sulfuric acid particles from the stratosphere (including both those with and without meteoric metals). The percentage contribution to each parameter in Figure 8 by stratospheric aerosol will be somewhat larger at ambient conditions because sulfuric acid has some water uptake even at <10% relative humidity. Ambient sulfuric acid particles may have roughly 5 to 15% larger diameters than measured in the warm aircraft cabin.

## 5.1 Infrared heating

An important property for stratospheric particles is their absorption and emission of infrared radiation. Infrared properties are more important for stratospheric aerosol than very low-altitude aerosol because the latter are close to the surface temperature and so absorb and emit similar amounts of energy.

Infrared absorption by stratospheric particles is important for two reasons. First, it heats the stratosphere around the particles. Changes in circulation due to infrared heating were responsible for significant changes in ozone after the Pinatubo eruption (Labitzke and McCormick, 1992; Pitari and Rizi, 1993). The heating-induced changes in ozone were as large or larger than those due to heterogeneous chemistry. There are additional feedbacks on the circulation after changes in infrared heating due to changes in ozone and water vapor (Visioni et al., 2017). Infrared heating is largest in the lower stratosphere where the temperature contrast with the surface is greatest (Lacis, 2015).

Second, infrared absorption by stratospheric particles offsets some or even all of the shortwave cooling of the Earth. For sulfuric acid particles similar to those after the Pinatubo eruption, longwave heating offset roughly 25% of the shortwave cooling (Hansen et al., 2005). This increases to about 50% for large injection rates because larger particles (>0.6 um) become

increasingly less efficient at scattering sunlight to outer space (Figure 7) compared to their volume (Niemeier and Timmreck, 2015). Large particles can even cause net warming: alumina with the size distribution from rocket emissions was calculated to cause net warming (Ross and Sheaffer, 2014).

For wavelengths much larger than the particles, absorption and emission are approximately proportional to total particle volume (and the material) and do not depend on particle size (van de Hulst, 1981). Therefore, net thermal infrared heating is insensitive to the size of particles and will approximately follow the volume distribution in the top panel of Figure 8. For ATom2, most of the volume distribution was composed of particles of stratospheric origin, but that was not always the case (Figure 1). The infrared effects of the tropospheric particles found in the lower stratosphere are hard to assess. Longwave

radiative heating depends not only on the strength of the absorption bands but also on their overlap with atmospheric windows in the infrared spectrum. The presence of a significant fraction of organic material has unknown implications for infrared heating. Like sulfate, infrared heating will be proportional to the volume of organic aerosol, but the absolute amount of heating depends on how aerosol absorption features correlate with gas-phase absorption or window regions in the infrared spectrum. We are not aware of any infrared spectra of organic material in aerosols in the stratosphere or upper troposphere.


**5.2 Scattering of visible light**

The middle panel in Figure 8 shows the size distribution weighted by the amount of sunlight scattered to outer space, which is relevant for shortwave climate effects. Weighting the size distribution by the light extinction would shift the peak just slightly

further to larger particles. Because of their size, the sulfuric acid particles contribute a greater fraction of the light scattering than their mass fraction. In fact, comparing Figures 1 and 7, the most abundant size of the sulfuric acid particles in the lower stratosphere was close to the maximum in light scattering to outer space per unit mass. The ATom mission took place during a time with small volcanic influence. In contrast, particles shortly after the Pinatubo eruption had volume mean diameters greater than 0.7 µm (Wilson et al., 2008; Figure S3), large enough that their mass scattering efficiency decreased. Figure S4

shows the relative extinction due to various particle types for cases other than the ATom2 Northern Hemisphere used as an example in Figure 8.

**5.3 Heterogeneous chemistry**

Particle size can affect heterogeneous chemistry. Reactions with sulfuric acid particles that are important to stratospheric chemistry span the range from reactions that occur in the interior of liquid particles and hence are proportional to volume to reactions that occur on the surface and hence are proportional to surface area (Hanson et al., 1994). Heterogeneous chemistry can be especially important within or at the edge of the polar vortex (Solomon et al., 2015; Stone et al., 2017).

The bottom panel of Figure 8 shows the size distribution weighted by surface area rather than volume. For these conditions surface reactions are closely proportional to surface area; gas-phase diffusion is a minor correction. For the case shown in Figure 8, the organic-sulfate particles from the troposphere are about half of the surface area in the lowermost stratosphere. This is significant because whereas stratospheric heterogeneous chemistry on sulfuric acid has been extensively studied, little is known about the same reactions on organic-sulfate particles. Figure 8 shows ATom2 Northern Hemisphere as an example.

Surface area fractions for other cases are shown in Figure S4.

    The organic-sulfate particles differ from the sulfuric acid particles in important ways. Most obviously, they contain a high proportion of organics that may participate in new chemistry with halogen radicals. Iodine in particular may react with organic aerosols (Murphy and Thomson, 2000; Pechtl et al., 2007). Although not fully neutralized, the organic-sulfate particles are not

nearly as acidic as the relatively pure sulfuric acid particles. This can be determined from acid cluster peaks in the PALMS mass spectra. Some chlorine activation reactions that lead to ozone destruction are acid-catalyzed (Burley and Johnston, 1992) and therefore may be slower on partially neutralized particles. The organic-sulfate particles also contain less water – sulfuric acid is extremely hygroscopic compared to other species at the low relative humidities in the stratosphere. The availability of condensed water for heterogeneous reactions could be further reduced if the organic-sulfate particles are glassy at the low

temperatures and humidity in the stratosphere (Krieger et al., 2012; Price et al., 2014).

### 5.4 Sedimentation

    Sedimentation is a key process in the stratospheric aerosol budget (Wilson et al., 2008). It is more important in the stratosphere

than it is near ground level partly because particles fall faster at lower air density. A bigger reason sedimentation is important in the stratosphere is the relevant time scale: a fall speed of a kilometer per month would be unimportant in the lower troposphere but can control the residence time of a particle in the stratosphere. Particles larger than about 1 μm diameter have sedimentation rates greater than 10 km $yr^{-1}$ in the lower stratosphere. The sedimentation flux as a function of size, shown in Figure S5, is similar to the light scattering panel in Figure 8). For the ATom2 example used in Figure 8, the sulfuric acid

particles have roughly twice the volume of the organic-sulfate particles. Their source strength must be somewhere between about twice as large as the tropospheric particles (if loss is controlled by bulk air motion) and three times as large (if loss is controlled by sedimentation).

### 5.5 Ultraviolet scattering and absorption


    Absorption and scattering of ultraviolet light are distinct from that of visible light because of the impact on photolysis rates. Light scattering by stratospheric aerosol changes the path length of light in the stratosphere, which in turn changes photolysis rates (Huang and Massie, 1997; Pitari et al., 2015). The calculations are complex because Rayleigh scattering in the ultraviolet

leads to strong effects from multiple scattering (Bian and Prather, 2002), especially at twilight or if a scattering aerosol layer is located above gas-phase absorption (Davies et al., 1993; Anderson et al., 1995). In addition, the long path lengths magnify the importance of any absorption of ultraviolet light by aerosols.

The scattering of ultraviolet light peaks at smaller particle sizes than for sunlight. This means that the smaller tropospheric-organic particles contribute substantially to the scattering of ultraviolet light. The relative contributions to scattering of light at < 240 nm are shown in Figure S5 and are similar to the surface area panel in Figure 8 except that sizes smaller about 80 nm and larger about 600 nm contribute less to UV scattering than they do to surface area.

One important wavelength band is 200 to 242 nm, where photolysis of $O_2$ is responsible for formation of ozone and photolysis of $N_2O$ produces odd nitrogen ($NO_y$). (Brasseur and Solomon, 1986). For purely scattering particles, changes in photolysis in this wavelength range are reduced by large cancellations in direct and diffuse light (Michelangeli et al., 1989). Light scattering by the El Chichon volcanic cloud was estimated to reduce $O_2$ photolysis by about 10% (Michelangeli et al., 1989). The overall effect of scattering seems to be to reduce ozone formation (Pitari et al., 2015).

Unlike pure scattering particles, absorbing particles would not have a similar partial cancellation between direct and diffuse sunlight. Huang and Massie (1997) examined the effect of substituting ash with visible and UV absorption for non-absorbing sulfuric acid in a simple model of photolysis after a volcanic eruption. There are competing effects on the ozone column because both $J_{O2}$ and $J_{O3}$ are reduced by UV absorption, with one reducing and the other increasing ozone (Pitari and Rizi, 1993). The individual effects were several percent of the ozone column with the net impact difficult to assess because their simple model did not include $NO_x$ or halogen chemistry. The imaginary refractive index of the organic-sulfate particles at wavelengths below 242 nm is not known but could easily be large enough to lead to significant absorption compared to scattering.

**6 Discussion**

We have shown in sections 3 and 4 that there are two distinct particle populations in the LMS, those of tropospheric origin and those of stratospheric origin, and that these have different size distributions. We have also shown in section 5 that these two populations, because of their different chemical composition, have different radiative efficiency and surface area per unit mass. These results motivate more general calculations about the radiative and chemical effects of particles as a function of size. Because we are concerned with the climate implications of stratospheric particles we will compare the sunlight scattered to outer space with various parameters such as surface area and sedimentation rate. The presence of more than one type of particle in the stratosphere also motivates investigating why different types of particles have different sizes.

## 6.1 Complex controls on particle size

The previous sections demonstrate that particle size is important for many of the properties relevant to climate and chemical effects. In the real world, one cannot instantly fill a box with monodisperse particles of a chosen size, the way one might in a model. It is important to understand what controls the size distribution of particles in the stratosphere and how it will change with additional aerosol or precursors such as sulfur dioxide. The mean particle diameter in the stratosphere is not a constant but varies with the aerosol loading, altitude, and latitude (English et al., 2012).


The size of particles in the unperturbed or perturbed stratosphere can be understood in two complementary ways. The first way, a top-down approach, says that for a given mass of stratospheric aerosol, the more particles there are, the smaller they must be. The second way, a bottom-up approach, considers how the size of each particle is set by a balance of growth and removal processes in the stratosphere.


For the top-down approach, one must consider at least three sources of particle number in the stratosphere. Particles come down from the mesosphere, up from the troposphere, and new particles can form in the stratosphere (Murphy et al., 2014). The meteoric source of particles to the stratosphere is mostly "smoke" consisting of material that evaporated from ablating meteoroids and condensed into new particles high in the atmosphere. Much of this material descends near the winter poles

(Bardeen et al., 2008). Second, tropospheric particles provide an important source of stratospheric particles below 20 km altitude (Yu et al., 2016). The fate of tropospheric particles entering the stratosphere is poorly represented in most models. New particle formation is also important for the stratosphere. The pure sulfuric acid category in Figure 1 is probably from growth of particles formed in (or at the edge of) the stratosphere. One formation region is near the tropical tropopause with upward transport into the stratosphere (Brock et al., 1995; English et al., 2011). There is probably also formation of new

sulfuric acid particles higher in the stratosphere over the winter poles (Wilson et al., 1989), although this must be distinguished from meteoric smoke descending in the same regions (Curtius et al., 2005).

Adding sulfuric acid or its precursors, either from volcanoes or potential intentional injection, will have complex effects on new particle formation, with more vapor to condense but also more surface area sink. In contrast, injected solid particles would

provide a surface sink for background sulfuric acid from oxidation of carbonyl sulfide, likely reducing new particle formation. This implies that injected solid particles would probably change the size of the natural sulfuric acid particles in the stratosphere. Sufficiently small injected solid particles might reach high altitudes where existing sulfuric acid particles have evaporated (Weisenstein et al., 2015; Jones et al., 2016). There could be unknown effects if they were later entrained in descending air in the winter polar regions.


The bottom-up approach considers how the size of stratospheric particles is determined by a balance of growth and removal processes. Particles grow by coagulation and by condensation of sulfuric acid and other species. Coagulation in the unperturbed stratosphere is slow except for special situations such as shortly after new particle formation (Brock et al., 1995; Hamill et al., 1997). Coagulation increases non-linearly with aerosol concentration so it becomes more significant after volcanic eruptions (Pinto et al., 1989) or large injection scenarios (Weisenstein et al., 2015). In these cases, coagulation helps drive the extra mass primarily to larger particles rather than more numerous particles (Heckendorn et al., 2009; Niemeier and Timmreck, 2015). Both sedimentation and downward motion are important removal processes (Wilson et al, 2008).

One implication of having multiple sources of particles in the stratosphere is that there is no single response to injected material. It is only in the last few years that stratospheric models have incorporated multiple sources of particles along with detailed microphysics (Pitari et al. 2014; Yu et al., 2016; Mills et al., 2017). There is still considerable uncertainty in quantitatively understanding the size of particles in the current stratosphere, let alone after a perturbation. Figure 2 demonstrates that a detailed microphysics model of the stratosphere did not grow the sulfuric acid particles to large enough sizes.

## 6.2 Impacts on the stratosphere as a function of size

The preceding section suggests that the ultimate size of particles is set by multiple processes and is not easy to predict. We therefore examine a range of particle sizes for calculations of the how the cooling of the Earth compares to other processes such as heating of the stratosphere and the potential for heterogeneous chemistry. Figure 9 shows an estimate of net cooling compared to particle volume, surface area, and sedimentation rate as a function of particle diameter in the lower stratosphere.

Because scattering has a much stronger size dependence than infrared absorption, the size that has the least infrared absorption for a given amount of shortwave climate impact is about 0.5 µm diameter (filled circles). This is true regardless of details of the infrared spectrum of the particles. Larger or smaller particles will be less effective at cooling the Earth and will cause more stratospheric circulation changes for a given amount of cooling. Sufficiently large (> 4 µm) or small (< 0.1 µm) particles cause net heating of the Earth (Lacis et al., 1992). The crossing point between net cooling and net heating will depend on particle composition. Even for the optimal size, the infrared heating due to deliberate injections of sulfuric acid or its precursors into the stratosphere would cause significant changes in circulation (Aquila et al., 2014).

The potential for increased heterogeneous chemistry would be reduced by using larger particles with less surface area. Particles with a diameter of about 1 µm have the largest cooling effect for a given surface area (open circles). These larger particles, however, have high sedimentation rates (downward triangles) compared to the amount of cooling they produce. For a given climate impact, the mass flux due to sedimentation is minimized by particles with a diameter of about 300 nm. The relative

sedimentation varies slightly with altitude but the pattern is similar. The least diffuse light is created by the smallest particles
(Figure 7).

**7 Summary**

The title of Solomon et al. (2011) includes the phrase "the persistently variable 'background' stratospheric aerosol". The ATom
data presented here add new meaning to that phrase. Out of eight samplings of the lowermost stratosphere, three exhibited
much higher aerosol concentrations for three different reasons: a volcanic eruption, biomass burning aerosol, and transport of
dust and other near-surface particles. None of these events were targeted during the flights. Such variations in the stratospheric
aerosol layer are important for both heterogeneous chemistry (Solomon et al., 1996) and climate (Solomon et al., 2011).

There are important differences in the aerosol in the lower stratosphere between the Northern and Southern Hemispheres. A
smaller amount of tropospheric aerosol in the Southern Hemisphere stratosphere indicates that the tropospheric particles are
mixing into the lower stratosphere within each hemisphere rather than entering in rising air in the tropics and splitting into the
two hemispheres. Sulfuric acid particles in the Southern Hemisphere also acquire less organic content. This suggests that there
are lower concentrations of gas-phase organics in the Southern Hemisphere. One of several possible formation routes is that
small organic compounds such as acetone and formaldehyde can react with concentrated sulfuric acid to form polymers that
stay in the aerosol (Iraci and Tolbert, 1997; Williams et al., 2010). Other routes would be if low-volatility organic molecules
were formed in the gas phase or evaporated from the tropospheric particles and recondensed on the sulfuric acid particles.
Even in the Northern Hemisphere, only low part-per-trillion range concentrations of gas-phase organics are required to explain
the very small amounts of organics taken up by the sulfuric acid particles.

The data here add support to the concept of Yu et al. (2016) that tropospheric particles comprise a significant fraction of the
aerosol in the lowermost stratosphere. Such tropospheric particles offer a route for anthropogenic influence on the stratosphere.
The Yu et al. model also correctly predicts that tropospheric particles are smaller than sulfuric acid particles formed in the
stratosphere (Figure 2).

The data here have several implications for satellite retrievals. First, the reason lower stratospheric size distributions are often
broad is that they are really two or more overlapping distributions. As the tropospheric/stratospheric mix shifts the mean
diameter will shift. Second, unlike sulfuric acid, the smaller, organic-rich particles coming up from the troposphere may be
effloresced and/or glassy. That could explain a very small amount of depolarization.  Finally, when only the stratospheric mode
is dominant the size distribution can be very narrow.

Absorption of ultraviolet light means that impurities should be considered when assessing deliberately added materials. For example, absorption appropriate for optical-quality sapphire should probably not be used when evaluating proposals to add industrial quantities of alumina to the stratosphere. Even part-per-million impurities in alumina increase absorption in the ultraviolet (Innocenzi et al., 1990). Compared to many materials, sulfuric acid has extremely low absorption in the ultraviolet (Noziere and Esteve, 2005; Dykema et al., 2016).

The broad distribution of particle sizes in the unperturbed stratosphere is the superposition of several narrower distributions. Single particle types, particularly meteoric-sulfuric acid particles, can have narrow size distributions (lognormal standard deviation ~1.4).

Multiple formation mechanisms for stratospheric particles imply that the size of particles after a volcanic or intentional injection may be difficult to predict. Yet an accurate prediction of size is important: The diameter must be known to perhaps 25% to accurately estimate tradeoffs between climate impact and side effects (Figure 9). A state-of-the-art microphysical bin model underestimates the size of stratospheric sulfuric acid particles, indicating that we do not fully understand what controls the size of particles in the stratosphere. The size difference has significant impacts on properties: the modeled particles have about 65% of the climate impact per unit mass as calculated from observed sizes of stratospheric sulfuric acid particles, 160% of the surface area, and sediment about 60% as fast.

There have also been numerous proposals for, and studies of, injecting material into the stratosphere for the purpose of solar radiation management (National Research Council, 2015). Regardless of the desirability of such actions, the calculations presented here on the optical properties and potential for heterogeneous chemistry have implications the impact of intentionally adding material to the stratosphere. There is no single diameter that produces the largest shortwave climate impact with the fewest side effects (Figure 9). To the extent that one could control the size of particles after an intentional injection, any chosen size involves tradeoffs. Particles smaller than about 0.6 μm diameter have more surface area for possible heterogeneous chemistry per unit cooling of the Earth. Particles larger than about 0.4 μm require more injected mass and produce more diffuse light. For a given amount of scattered sunlight, either sufficiently large or small particles have more infrared absorption and hence more impacts on stratospheric circulation. Most of the mass of particles after the Mt. Pinatubo eruption was larger than 0.6 μm diameter (Brock et al., 1993; Wilson et al. 2008), a size range with relatively little surface area compared to their climate impact. The heterogeneous chemistry observed after Mt. Pinatubo may therefore underestimate what might happen with intentionally added material.

**Data availability and author contributions**

Data are available at https://daac.ornl.gov/ATOM/campaign/, https://espo.nasa.gov/atom/content/ATom, and
https://www.esrl.noaa.gov/csl/projects/atom/data/MurphyACP2020/. DMM wrote the manuscript with assistance from all
other authors. DMM, KDF, CAB, AK, GPS, and CJW made aerosol measurements and analysed the data. IB and CRT made
ozone measurements and analysed the data. PY performed model calculations. Authors declare no competing interests.

**Acknowledgements**

T. Ryerson helped provide ozone measurements. J. C. Wilson provided data from his 2008 paper. These measurements were
supported by NOAA internal climate funding and also in part by NASA award NNH15AB12I. The ATom mission as a whole
was supported by NASA's Earth System Science Pathfinder Program EVS-2 funding. A. Kupc was supported by the Austrian
Science Fund's Erwin Schrodinger Fellowship J-3613.

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

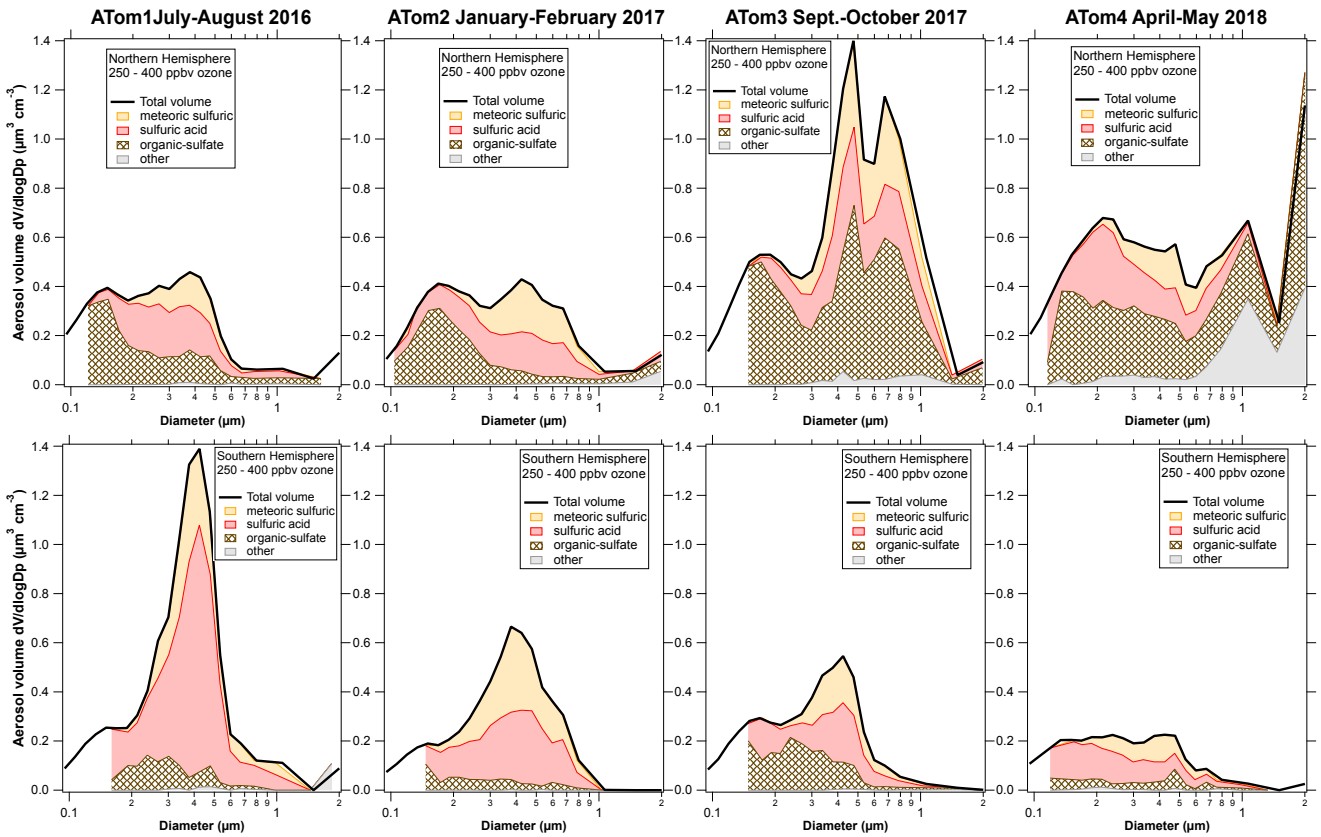

**Figure 1. Composition resolved size distributions from the lower stratosphere during the Atmospheric Tomography (ATom) mission. The size distributions from optical particle counter measurements are apportioned at each size into classes of particles based on the PALMS single particle composition data. The smallest few size bins sometimes had too few particles with PALMS mass spectra to apportion the composition. Sizes in all figures are geometric diameter at low relative humidity and concentrations are per standard cm³. Fine structure in some total volume distributions at 0.6 μm and larger may be artifacts due to Mie resonances in the optical particle counter; minima near 0.2 or 0.3 μm are robust.**

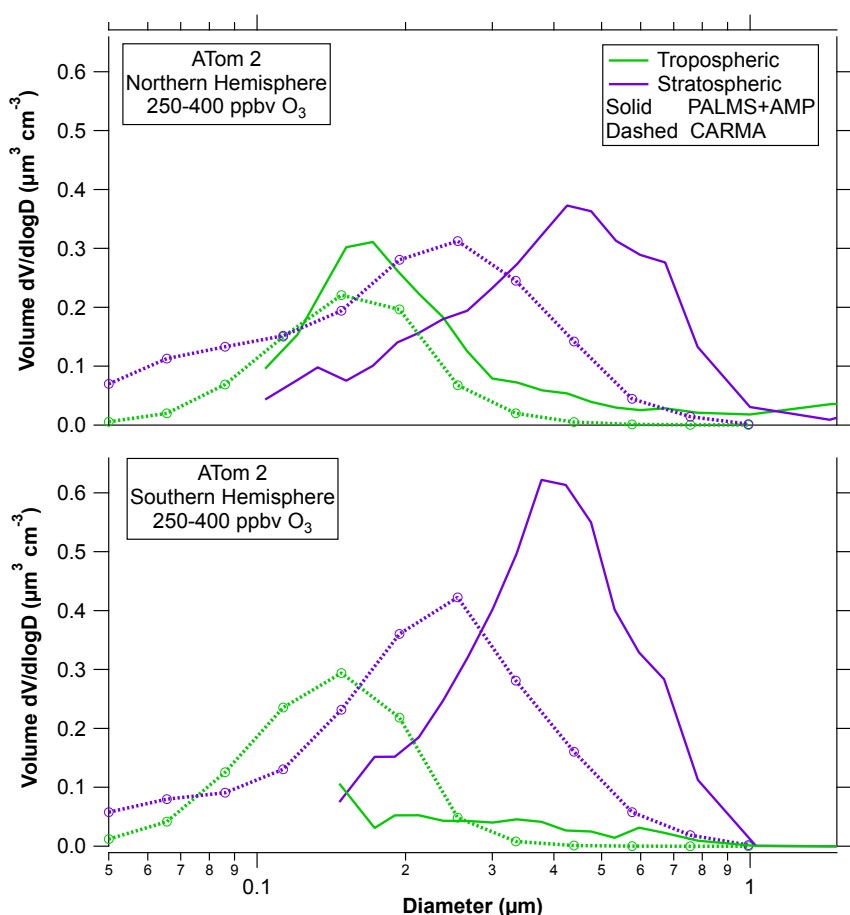

**Figure 2. A comparison of the CARMA bin microphysics aerosol model in the CESM climate model with observed size distributions during ATom2. Total particle volumes are proportional to areas under each curve. The PALMS stratospheric particles are the sulfuric acid and meteoric-sulfuric acid types. Tropospheric particles are mixed organic-sulfate particles and less common other types. The model tracks stratospheric and tropospheric particles separately.**

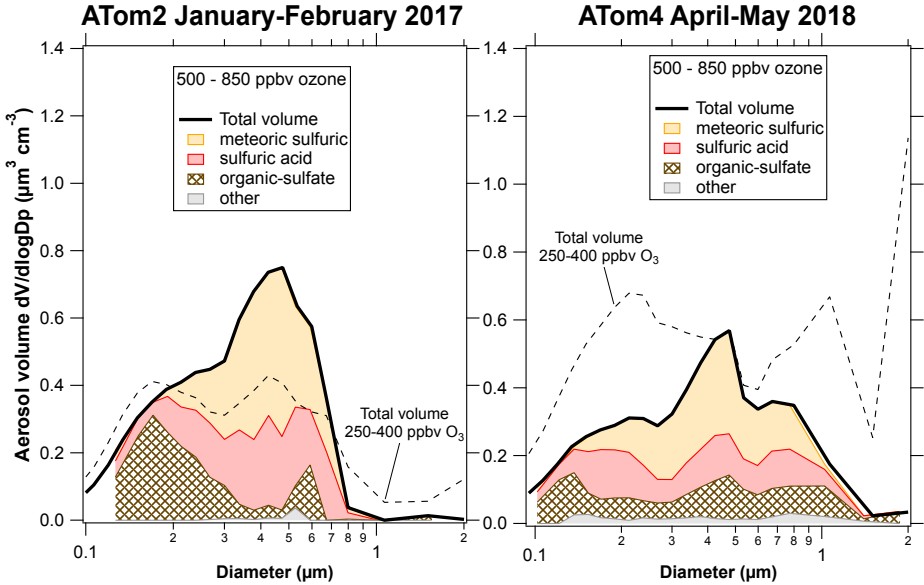


**Figure 3. Composition-resolved size distributions for the times when data are available for ozone greater than 500 ppbv, all in the Northern Hemisphere. The dashed curves are to facilitate comparison with Figure 1.**

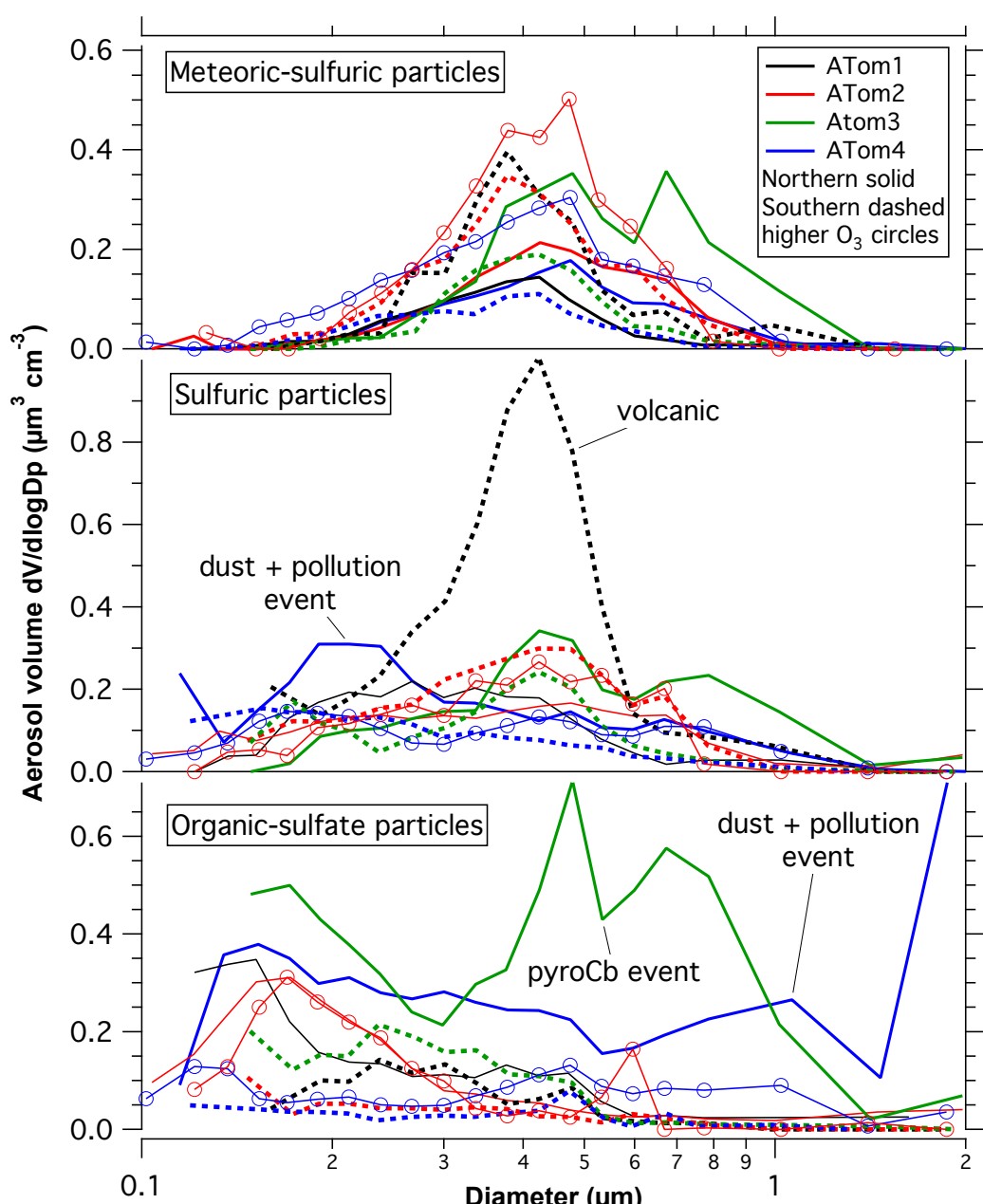


**Figure 4. Size distributions of several types of particles in the lower stratosphere. Northern Hemisphere data are solid and Southern Hemisphere data are dashed. Data are from 250 to 400 ppbv ozone except the curves with circles are data at more than 500 ppbv ozone. The three major events are noted where they may have influenced the size distributions. Lognormal fits are given in Table 1.**


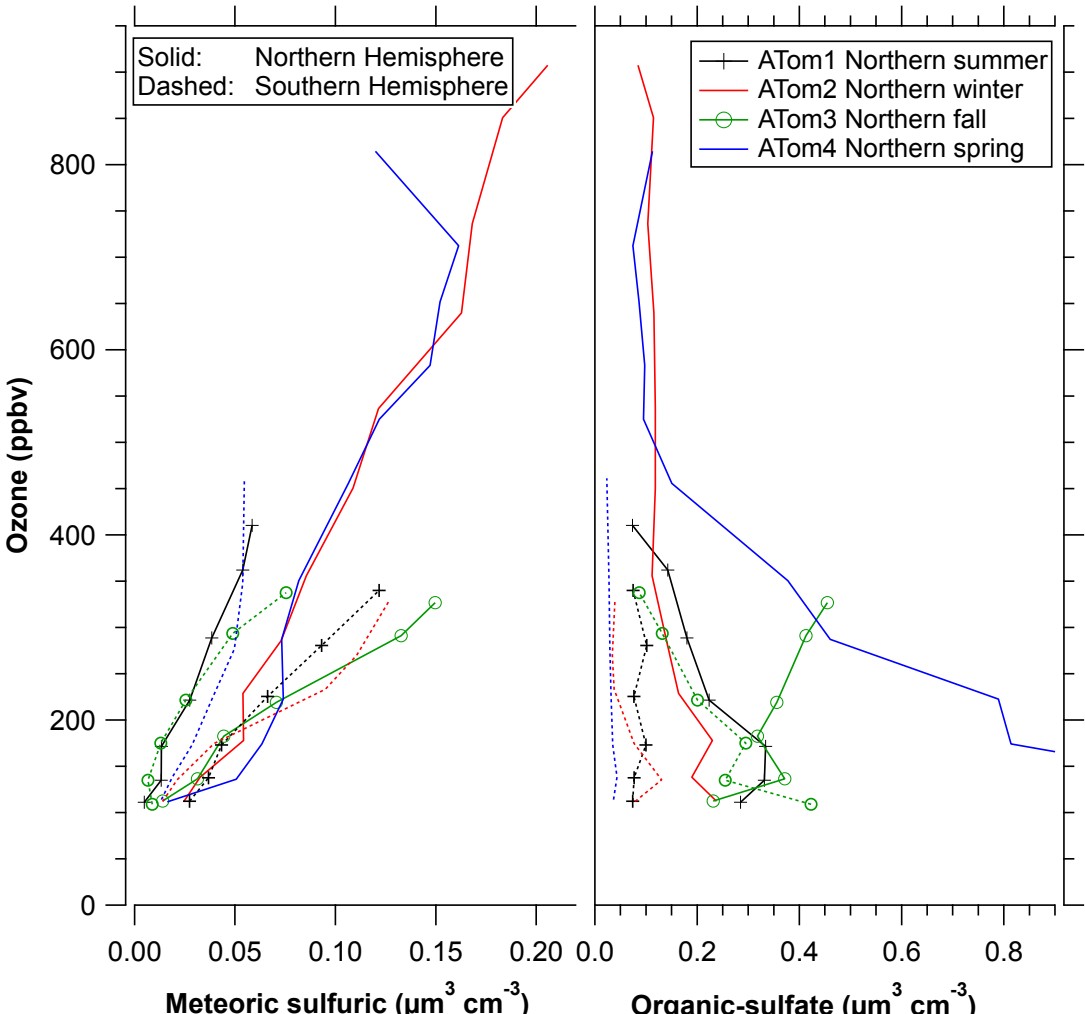

**Figure 5. Profiles of the volume concentration of two types of particles in the lower stratosphere, using ozone as a surrogate for a vertical coordinate. This does not include particles smaller than 0.1 µm. Sulfuric acid particles with meteoric metals have a high-altitude source whereas organic-sulfate particles, as selected by their mass spectra, originate in the troposphere. The organic-sulfate particles are more abundant in the Northern Hemisphere than they are in the Southern Hemisphere.**

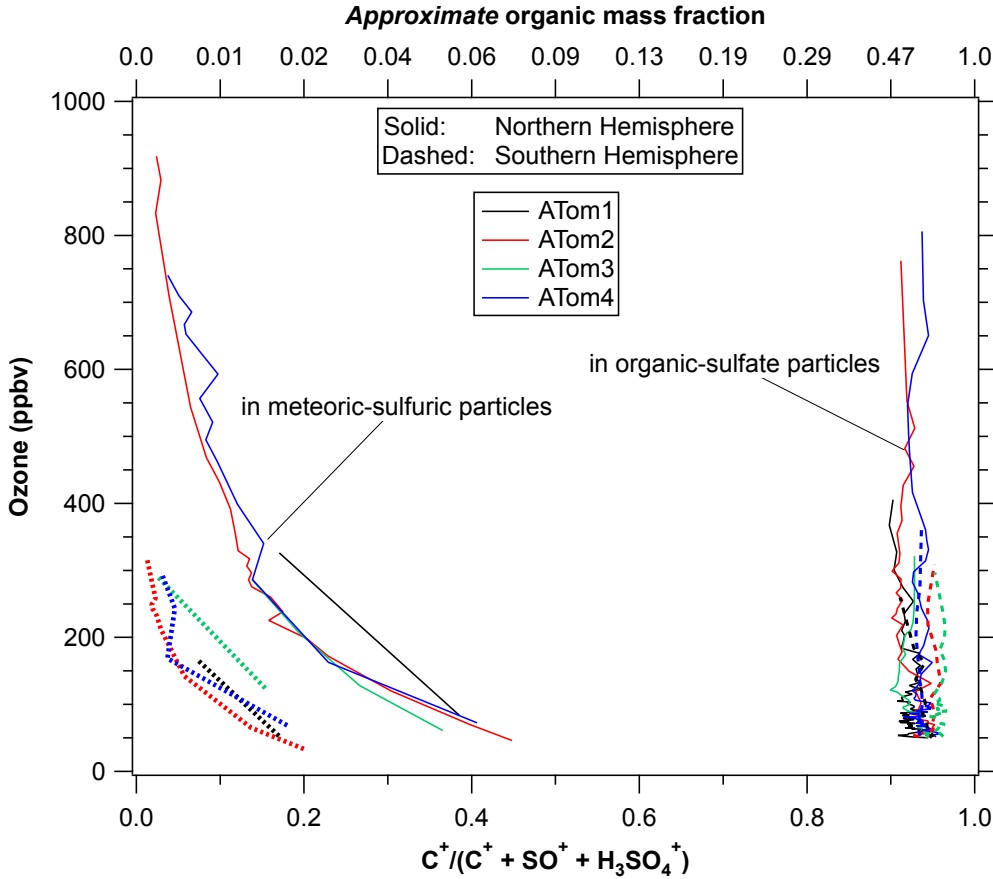


**Figure 6. The organic content of two types of stratospheric particles using ozone as a surrogate for altitude. Shown are the relative areas of the C$^+$ peak in the mass spectra relative to two peaks indicative of sulfate. The top axis is an approximate mass fraction. The organic-sulfate particles are about 40 to 70% organic by mass, consistent with a**

**tropospheric source. The sulfuric acid particles have near-zero organic content well into the stratosphere, increasing up to a few percent by mass near the tropopause. There was a strong difference in the organic content of the sulfuric acid particles between the hemispheres. Each point is the average of about 200 mass spectra.**

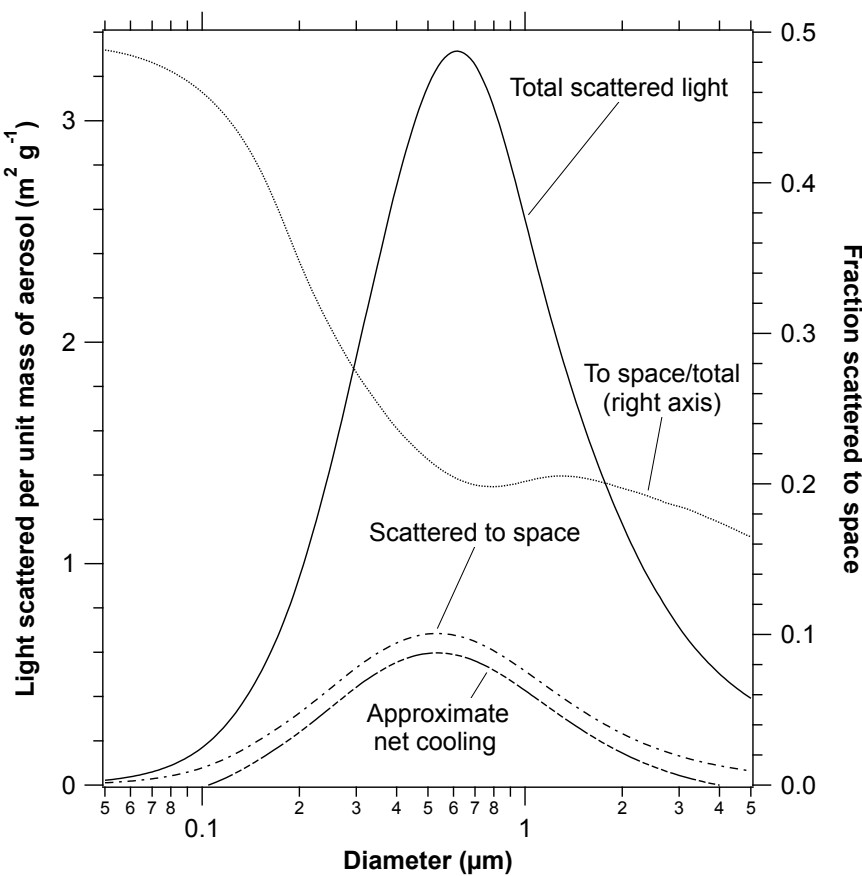


**Figure 7. The mass scattering efficiency of particles with a refractive index similar to sulfuric acid calculated from Mie scattering. Shown are the total scattered light, the light scattered away from the Earth to space, and their ratio. Larger particles have more forward scattering and therefore a smaller fraction scattered to space. The lowest curve shows the climate impact after an approximate correction for warming due to infrared absorption. The infrared heating is the**

**difference between the two lower dashed curves. Particles smaller than about 0.1 μm or larger than about 4 μm warm the Earth (Lacis et al., 1992). Tropospheric particles containing ammonium sulfate or other substances with higher refractive indices would shift the curves to slightly larger diameters.**

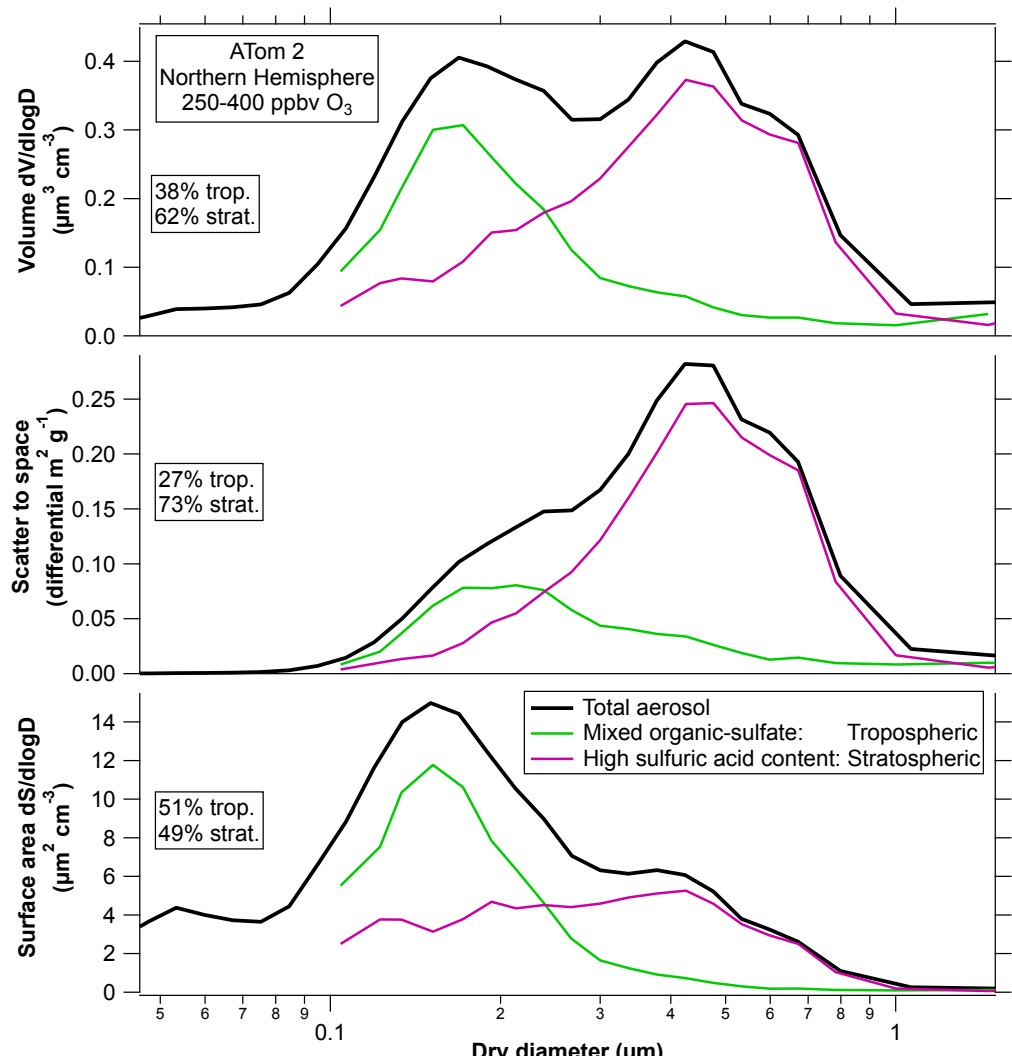


**Figure 8. The composition-resolved size distribution from Figure 1 for the ATom2 Northern Hemisphere weighted by volume, light scattered to outer space, and surface area. The two sulfuric acid categories (with and without meteoric metals) have been combined. Percentages refer to sizes above 0.1 μm only. ATom2 is chosen as an example; Figure S4 shows the percentage contributions of various particle types to these processes for other deployments and for the**

**Southern Hemisphere. Scaling to net thermal infrared heating gives nearly identical relative contributions as volume in the top panel. Scaling to sedimentation rate gives a similar shape to the middle panel.**

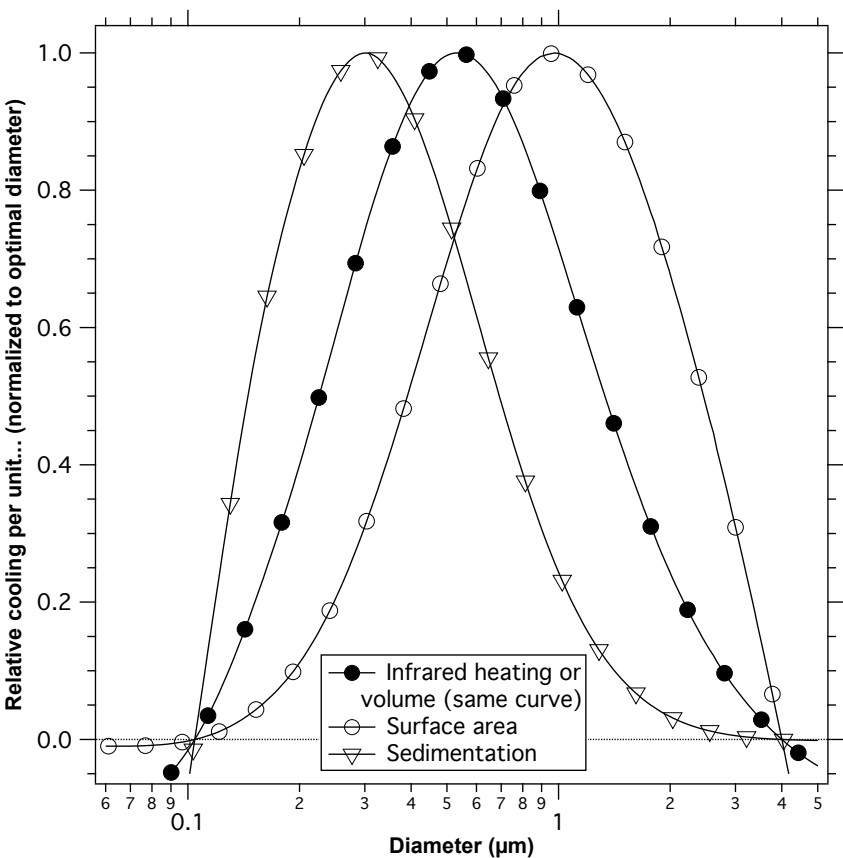

**Figure 9. Calculated net cooling per unit volume, surface area, and sedimentation velocity. Each curve is normalized**
**so the maximum point is unity. For example, 0.2 μm particles have about one tenth the net cooling per unit surface area**
**as 1 μm diameter particles. Particles of about 0.5 μm diameter give the most cooling per unit mass and the most cooling**
**per amount of infrared heating of the stratosphere (volume and infrared heating have the same normalized curves).**
**Particles of about 0.3 μm diameter give the most cooling per unit sedimentation rate. Calculations of sunlight reflected**
**to outer space are for a single size aerosol averaged over the solar spectrum and zenith angles characteristic of the**
**sunlit Earth. The wavelength averaging eliminates the oscillations from Mie scattering.**

| | Total Volume | | Meteoric-sulfuric | | | Sulfuric | | | Organic-sulfate | | | Other |
|---|---|---|---|---|---|---|---|---|---|---|---|---|
| | 6 nm-1.7 µm | 0.1-1.7 µm | Volume | Fit diam. | Fit σ | Volume | Fit diam. | Fit σ | Volume | Fit diam. | Fit σ | Volume |
| | µm³ cm⁻³ | µm³ cm⁻³ | µm³ cm⁻³ | µm | | µm³ cm⁻³ | µm | | µm³ cm⁻³ | µm | | µm³ cm⁻³ |
| ATom1 NH | 0.35 | 0.30 | 0.046 | 0.37 | 1.35 | 0.103 | 0.29 | 1.55 | 0.150 | 0.13 | 1.89 | 0.004 |
| ATom2 NH | 0.35 | 0.33 | 0.083 | 0.46 | 1.44 | 0.115 | 0.35 | 2.14 | 0.124 | 0.17 | 1.45 | 0.005 |
| ATom3 NH | 0.78 | 0.76 | 0.150 | 0.54 | 1.51 | 0.160 | 0.50 | 1.74 | 0.436 | *0.39* | *2.89* | 0.024 |
| Atom4 NH | 0.68 | 0.64 | 0.066 | 0.44 | 1.49 | 0.175 | *0.22* | *2.15* | 0.292 | *0.30* | *2.04* | 0.107 |
| ATom1 SH | 0.49 | 0.48 | 0.101 | 0.39 | 1.27 | 0.330 | 0.40 | 1.29 | 0.050 | 0.27 | 1.43 | 0.011 |
| ATom2 SH | 0.32 | 0.31 | 0.110 | 0.40 | 1.39 | 0.147 | 0.39 | 1.64 | 0.052 | 0.12 | 1.80 | 0.001 |
| ATom3 SH | 0.28 | 0.26 | 0.056 | 0.40 | 1.29 | 0.088 | 0.35 | 1.73 | 0.120 | 0.19 | 1.93 | 0.003 |
| ATom4 SH | 0.19 | 0.17 | 0.040 | 0.36 | 1.47 | 0.090 | 0.16 | 2.30 | 0.033 | *0.20* | *2.94* | 0.004 |
| ATom2 NH high | 0.40 | 0.39 | 0.156 | 0.42 | 1.36 | 0.122 | 0.40 | 1.64 | 0.112 | 0.18 | 1.36 | 0.005 |
| ATom4 NH high | 0.37 | 0.35 | 0.132 | 0.41 | 1.59 | 0.104 | 0.31 | 2.57 | 0.093 | *0.38* | *3.64* | 0.021 |

**Table 1. Total volumes and volumes and log-normal fit parameters for individual types of particles. Lognormal fits were performed for particle volume so the fit diameter is for the volume distribution. The volumes and fits are for the size range 0.1 to 1.7 µm diameter except as noted (first column). Data are for 250-400 ppbv ozone except the last two rows are for 500-850 ppbv. Volumes are per standard cm³. Fits are to the equation $V = a*\exp(-(\ln D - \ln D_{fit})^2/(2\sigma^2))$ where $D$ is the diameter and $a$ is a scaling factor. Each fit is for a specific component. Fits were not attempted for "other" particles (mostly dust) because they did not generally show a defined mode. Italics indicate a poor fit to a single lognormal shape.**