# Peer review of "Radiative and chemical implications of the size and composition of aerosol particles in the existing or modified global stratosphere"

_Atmospheric Chemistry and Physics, 2020_

## Referee Comment (RC1) · Anonymous Referee #1 · 22 Oct 2020

The manuscript "Radiative and chemical implications of the size and composition of aerosol particles in the existing or modified global stratosphere" by Murphy et al., describes measurements of stratospheric size resolved aerosol composition from optical particle counters and the PALMS aerosol mass spectrometer flown on the ATom aircraft missions in 2016-2018. The authors discuss the compositional signatures that could be indicative of aerosol transport and formation pathways that yield aerosol with two distinctly different size modes. The radiative implications of these two distinct size modes are discussed and finally the implications of these for future geoengineering. Significant anomalies in stratospheric aerosol loading and composition that are associated with volcanic eruptions, stratospheric smoke injections and transport of dust into

the stratosphere are mentioned but not discussed in detail. Efforts to simulate the observed stratospheric aerosol size distributions using the CESM/CARMA model are briefly described.

The measurements of size resolved composition of lower stratospheric aerosol described in this work are unique, novel and of significant scientific importance. As this manuscript correctly describes, these results have broad implications for stratospheric chemistry and radiation and thus climate. The authors present an overview of the composition measurements in sections 1-4 at a high level that provides a brief description of the inherently complex topic of size dependent aerosol composition in the UTLS. However, the superficial description and analysis of the data set presented in this manuscript is insufficient to support the broad and generalized observations and conclusions presented in Sections 5 and 6. Many of these observations and conclusion are likely to be correct, and may be supported by this ATom data set, but have not been demonstrated to the reader in this work.

As this appears to be the first description of the PALMS composition measurements from the ATom campaign, the brief and qualitative description of the sampling and do not allow for the reader to understand how representative or significant the compositional analysis is, how definitively tropospheric and stratospheric air masses are separated, or even basic information such as how the ozone measurements were acquired. While the authors state that about 10,000 particles were sampled in the Northern Hemisphere (line 238) there is no indication of how these particles were divided between the various flights, how many flights there were, at what locations these particles were sampled, what fraction of the data was from tropopause folds, or how many particles were measured in the Southern Hemisphere. Figure S1 provides some insight into where the measurements may have been made, but as the color bar in S1 does not correspond with the 250 ppbV cut off for stratospheric air, even this provides insufficient information. Due to the low ceiling of the DC-8, the authors indicate that the sampled "stratospheric air was always associated with low tropopauses,

sometimes in tropopause folds". Given that the upper tropospheric organic aerosol loading can be very substantial, particularly with the wild fire activity likely present for ATom1 and ATom3, it is important to know whether these data are from stratospheric measurements above the broader tropopause (less likely in the summer time) or from tropopause folds (more likely in the summer time) and how edge effects and mixing were considered for the latter.

Without the presentation of a more thorough analysis of these data, it is not clear that some of the major conclusions of the paper are solidly supported. It may be the case that the transport of tropospheric organic-sulfate aerosols is producing a second smaller stratospheric aerosol mode centered around 200 nm in some specific cases. But this is not robustly supported in 5 of the 8 cases show in Figure 1. In ATom4 NH, the peak near 200 nm appears to primarily driven by an increase in pure sulfuric acid aerosol. In only one of the SH cases (ATom 3) is there convincing evidence of a secondary mode, and again in this case it appears to be driven by sulfuric acid aerosol (presumably from Calbuco). Without much more detailed description and analysis it is hard to be sure that the conclusion that tropospheric aerosol significant to the global lower stratosphere (paragraph starting on line 450) are really supported by this data set. This would be a very important result and I encourage the authors to layout a more convincing case based on the ATom data set.

In section 4, the authors state "As expected for their sources, the concentration of meteoric-sulfuric particles increases with altitude and the concentration of organic-sulfate particles decreases with altitude." While it does appear to be clear that concentration of meteoric-sulfuric aerosol increases with ozone/altitude, the data presented here do not appear to provide convincing support for a general decrease in organic-sulfate particles with altitude. This relationship appears to be fundamental to apportioning the source of these particles to transport across the tropopause. Three of the organic-sulfate profiles shown in Figure 4 appear to be constant with respect to ozone/altitude, three may show a decrease and one shows an increase. The significance of these slopes is hard to determine without some metric of the measurement uncertainty. From the narrative, it appears that there was a limited (but unspecified) amount of particle composition data at ozone levels above 500 ppbV, so an indication of how significant the measurements > 500 ppbV is necessary to interpret the importance of this slope. Comparing the ATom 2 data in Fig 3 with Fig 1, it appears that there is in fact no significant difference in organic-sulfate particles in the 200nm mode, and actually a significant increase in the amount of organic-sulfate in the 600nm mode. The tropospheric origin of these organic-sulfate particles could also be indicated by their compositional signature, as the authors state on line 143. This is not demonstrated to the reader, even though the data must surely be available.

Throughout the paper there is no quantitative discussion of the uncertainty or statistics associated with this highly averaged data (in the narrative or in any of the figures). This issue is addressed anecdotally when convenient. For example, in the caption in figure 1 the authors do state that there 'may be artifacts' due to Mie resonance for certain features but other features are robust without any further explanation or justification. If there were other publications describing this data set in detail, some of these considerations could be addressed through references to these publications, however it appears that these publications are planned for the future.

Section 5 of the manuscript primarily describes the radiative and chemical impact that arises from the divergent size modes between aerosol of tropospheric and stratospheric origin. The narrative in section 5 is somewhat disjointed and difficult to follow as it is not always clear how it relates back to the measurements presented in the first sections of the paper. For instance, in section 5.1, the infrared absorption is only discussed in the context of particle volume, yet while the authors acknowledge that particle composition plays a role (line 266) this is not tied back to one of the primary observational findings, that the two size modes have significantly different composition. There are also several results that described in section 5 that are not demonstrated to the reader. For example, on line 329: "The relative contributions to scattering of light at < 240 nm

are fairly similar to the surface area panel in Figure 7 except that sizes smaller about 80 nm and larger about 600nm contribute less to UV scattering than they do to surface area." While this statement is quite likely true, it has not been shown and it is hard to quantify what 'fairly similar' describes. A similar statement begins on line 313 "On Figure 7, the sedimentation flux as a function of size would be slightly more skewed to large diameters than the light scattering panel." Such generalized statements need to be shown to be supported by the data.

Section 6 of the manuscript sets out to describe the relevance of this work to volcanic eruptions or to future geoengineering projects. While the discussion is interesting it is highly speculative and largely unrelated to the data presented in Sections 1-4. For example, in the paragraph starting on line 396 it is implied that differences between the modeled sulfuric acid particle sizes and these measurements (Figure 2) is a consequence of one of the main results that is presumably shown by the data – multiple sources of stratospheric aerosol. However, no details on the initialization for the model run or analysis of what is driving these discrepancies is provided, as a result the comparison in Figure 2 is largely anecdotal.

Finally, the title of the paper "Radiative and chemical implications of the size and composition of aerosol particles in the existing or modified global stratosphere" is misleading. While the aerosol composition data is presented in a limited way, the implications (described in sections 5 and 6) are based entirely on the size distribution of the particles not their composition, even when composition would certainly be important to these implications (e.g. 5.2 Scattering and 5.3 chemistry).

Technical Comments:

Line 54: "The moles of oxygen were approximately four times sulfur plus about 0.2 times carbon" - rephrase to make it clear where the brackets are $O2 = 4 * (S + 0.2C)$?

Paragraph starting line 95: It is unclear why the OPC data is needed if PALMs provides sizing?

Line 174: Using ozone as a proxy for altitude is understandable, but some indication of what an equivalent tropopause relative altitude or range of altitudes that a given ozone mixing ratio corresponds with would be useful.

Figure 6: What is the unit of measure for 'Approximate net cooling' and how was this calculated?

Figure 7: The units of these plots are unclear. Either provide units, or normalize the data to make it clear that it is relative surface/scatter/volume

Figure 8: This figure is confusing. What is meant by the IR-heating trace? Is the peak at 0.5 the most relative cooling per IR heating, and would that mean the least or most IR heating?

Figure S1: It would be useful to start the color bar at 250ppbv so that only the points used in this study are highlighted, with a clear distinction for where the particles shown in Figure 3 were found.

―――――――――――――――――――

---

## Referee Comment (RC2) · Anonymous Referee #2 · 4 Nov 2020

The LMS is an important part of the stratosphere and difficult to observe. Its composition differs from the rest of the stratosphere in many ways due to the mixing of tropospheric and stratospheric air. The study by Murphy et al. is based on in-situ data from aircraft of aerosol particle size distributions and composition of individual particles. Particles were classified depending on their composition to study the history of the particles. Radiative impact was discussed in relation to volcanic eruptions and climate engineering.

I find large issues with the authors' interpretation of the data. One reason is the generalization in sections 5-7 based on few data (a single season in a single year). Another

is the data on organics, and the claim that the organic aerosol comes from the tropo-
sphere only, which contradicts previous work. No uncertainties are presented for the
observations as far as I can tell, and there are no statistical analysis to support the
claims of trends.

I find several shortcomings in the manuscript and cannot recommend publication with-
out changes in the data analysis / interpretation of the data.

Major comments

1. Uncertainties and quantification a. How large are the uncertainties in the observa-
tions? I don't find any numbers on that. b. How well can one quantify different aerosol
constituents from the PALMS data? c. Is it possible to tell whether there is a trend, as
in Figure 4, if there are no statistical analysis to support it?

2. There are too little organics in the data compared to previous work, and this is not
discussed in relation to those studies. a. There is no discussion on organics after Cal-
buco. Several other groups have found large amounts of organics in volcanic aerosol.
Two examples are Schmale 2010 and Andersson 2013 reported much higher organ-
ics (or carbon) to sulfate ratios in volcanic particles. Those data were from AMS and
Ion beam analysis. b. Vertical gradient in organics is different from Martinsson 2019,
who reported higher carbon abundance deeper into the LMS. I cite from their abstract:
"...the carbonaceous and sulfurous components of the aerosol in the lowermost strato-
sphere (LMS) show strong increases in concentration connected with springtime sub-
sidence from overlying stratospheric layers. The LMS concentrations significantly
exceed those in the troposphere, thus clearly indicating a stratospheric production of
not only the well-established sulfurous aerosol, but also a considerable but less under-
stood carbonaceous component..." c. There is no real discussion on wildfire smoke.
The smoke from the Aug 2017 fires in western North America is evident in Figure 4.
The relatively small impact on the organics after the wildfire in that figure is strange
given that the event was almost volcanic sized (Peterson 2018). In Figure 1, it is apparently less organics 1-2 months after the fire than in spring the next year. Why is that? The discrepancy between the data in the manuscript with data from other studies is not discussed as far as I can tell. Identifying particles containing organics is not the same as measuring the mass (of organics or carbon), which other techniques do.

3. Large part of the manuscript is focused on a single season in the Northern Hemisphere (Atom2). It is unclear to me why this is the case. Sulfate concentrations in the LMS varies with season due to both seasonal variation in subsidence from the stratospheric overworld and varying cross TP transport. Stratospheric influence is large in winter and spring, and low in summer and fall. Thus: a. Data presented for a specific season is representative for that season only, and not for the entire year. b. General climatic conclusions cannot be drawn from a single season in a single year.

4. Data after the Calbuco eruption are almost not discussed at all. The authors mention the eruption and that it had some impact on the sulfuric acid but no more details. I understand that the authors focused on the Northern Hemisphere, but this omission is strange to me. I expect some discussion on data that are included in a manuscript.

Other comments

1. I would like to see a more clear discussion on the history of sulfuric particles. Meteoric ones from the upper stratosphere, and pure sulfuric from the lowest stratosphere (directly from the tropical lower stratosphere in the BDC).

2. The ExTL has very different composition than the rest of the LMS and (stratosphere). The author never mention the ExTL. Why is that?

3. I think that the phrase biomass burning shall be changed to wildfire smoke since it comes from PyroCb intrusion(s) to the stratosphere. Biomass burning leads the reader to believe that it is a general upwelling from diffuse fires instead of PyroCb formation in enormous fires.

4. I think that the manuscript should have a concise conclusions section after the

discussions section.

L45. "...The local tropopause and slightly above the altitude of the tropical tropopause...". This is not true. The tropical TP is located at ∼17 km, and the LMS extends to ∼14-15 km in the extratropics. L135. "...The primary source of sulfuric acid in the stratosphere, oxidation of carbonyl sulfide, is similar in the two hemispheres...". Do you mean the background stratospheric aerosol? It is still an open question whether SO2 or COS is the most important for the background stratospheric aerosol. L183. Change to: ...Northern hemisphere than in the Southern Hemisphere... L191. I do not understand the statement: "...the organic-sulfate particles contain on average about 40 to 80 organic material by mass..." L260. "...Alumina with the size distribution from rocket emissions was calculated to cause net warming (Ross and Sheaffer, 2014)..." This is a strange comment in the end of a section on volcanic particles and IR. Are there a sentence missing? L361. Change "will it" to "it will" L469. "...the modeled particles have about 65% of the climate impact per unit mass as the observations, 160% of the surface area, and sediment about 60% as fast..." Is this referenced to findings in the manuscript? Are there actual observations of these quantities or do you mean estimated from the PALMS observations?

---

## Author Comment (AC1) · 30 Dec 2020

Reply to reviewers, "Radiative and chemical implications of the size and composition of aerosol particles in the existing or modified global stratosphere" by Daniel M. Murphy et al. We thank the reviewers for their detailed comments that have improved the manuscript.

The comments are numerous so we are putting replies in-line with the comments. Since some formatting is lost when posted, our replies start with a dash in order to distinguish them from the reviewers' comments.

Anonymous Referee #1

 The manuscript "Radiative and chemical implications of the size and composition of aerosol particles in the existing or modified global stratosphere" by Murphy et al., describes measurements of stratospheric size resolved aerosol composition from optical particle counters and the PALMS aerosol mass spectrometer flown on the ATom aircraft missions in 2016-2018. The authors discuss the compositional signatures that could be indicative of aerosol transport and formation pathways that yield aerosol with two distinctly different size modes. The radiative implications of these two distinct size modes are discussed and finally the implications of these for future geoengineering. Significant anomalies in stratospheric aerosol loading and composition that are associated with volcanic eruptions, stratospheric smoke injections and transport of dust into the stratosphere are mentioned but not discussed in detail. Efforts to simulate the observed stratospheric aerosol size distributions using the CESM/CARMA model are briefly described. The measurements of size resolved composition of lower stratospheric aerosol described in this work are unique, novel and of significant scientific importance. As this manuscript correctly describes, these results have broad implications for stratospheric chemistry and radiation and thus climate. The authors present an overview of the composition measurements in sections 1-4 at a high level that provides a brief description of the inherently complex topic of size dependent aerosol composition in the UTLS. However, the superficial description and analysis of the data set presented in this manuscript is insufficient to support the broad and generalized observations and conclusions presented in Sections 5 and 6. Many of these observations and conclusion are likely to be correct, and may be supported by this ATom data set, but have not been demonstrated to the reader in this work.

- We appreciate the sentiment that the measurements are "unique, novel, and of significant scientific importance."

As this appears to be the first description of the PALMS composition measurements

from the ATom campaign, the brief and qualitative description of the sampling and do not allow for the reader to understand how representative or significant the compositional analysis is, how definitively tropospheric and stratospheric air masses are separated, or even basic information such as how the ozone measurements were acquired. While the authors state that about 10,000 particles were sampled in the Northern Hemisphere (line 238) there is no indication of how these particles were divided between the various flights, how many flights there were, at what locations these particles were sampled, what fraction of the data was from tropopause folds, or how many particles were measured in the Southern Hemisphere. Figure S1 provides some insight into where the measurements may have been made, but as the color bar in S1 does not correspond with the 250 ppbV cut off for stratospheric air, even this provides insufficient information. Due to the low ceiling of the DC-8, the authors indicate that the sampled "stratospheric air was always associated with low tropopauses, sometimes in tropopause folds". Given that the upper tropospheric organic aerosol loading can be very substantial, particularly with the wild fire activity likely present for ATom1 and ATom3, it is important to know whether these data are from stratospheric measurements above the broader tropopause (less likely in the summer time) or from tropopause folds (more likely in the summer time) and how edge effects and mixing were considered for the latter.

- We have added Table S.1 of when and where the DC8 was in the stratosphere and the relevant number of mass spectra acquired. A citation has been added for the ozone instrument. The way edge effects and mixing for tropopause folds was considered is largely through the use of ozone as a criterion: excessive mixing with tropospheric air will reduce the ozone levels below 250 ppbv. This is now more clearly stated. We have added tropopauses to Figure S1; they show that most of the data with more than 250 ppbv of ozone in all seasons is above a high latitude tropopause. We have changed the color scale on Figure S1 that starts at 250 ppbv.

Without the presentation of a more thorough analysis of these data, it is not clear

that some of the major conclusions of the paper are solidly supported. It may be the case that the transport of tropospheric organic-sulfate aerosols is producing a second smaller stratospheric aerosol mode centered around 200 nm in some specific cases. But this is not robustly supported in 5 of the 8 cases show in Figure 1. In ATom4 NH, the peak near 200 nm appears to primarily driven by an increase in pure sulfuric acid aerosol. In only one of the SH cases (ATom 3) is there convincing evidence of a secondary mode, and again in this case it appears to be driven by sulfuric acid aerosol (presumably from Calbuco). Without much more detailed description and analysis it is hard to be sure that the conclusion that tropospheric aerosol significant to the global lower stratosphere (paragraph starting on line 450) are really supported by this data set. This would be a very important result and I encourage the authors to layout a more convincing case based on the ATom data set.

- The referee's comment about not seeing that tropospheric particles are smaller shows that the size distributions weren't all that visible in the stacked plots of Figures 1 and 3. They also needed additional explanation. We've added a new figure (4) and associated text that show this more clearly. We have also added text (near new line 230) saying that the NH ATom4 case noted by the reviewer was in a dust event with strong transport from near-surface sources of particle other than dust. We appreciate the reviewer pointing out something that may not have been obvious to readers.

In section 4, the authors state "As expected for their sources, the concentration of meteoric-sulfuric particles increases with altitude and the concentration of organic- sulfate particles decreases with altitude." While it does appear to be clear that concentration of meteoric-sulfuric aerosol increases with ozone/altitude, the data presented here do not appear to provide convincing support for a general decrease in organic- sulfate particles with altitude. This relationship appears to be fundamental to apportioning the source of these particles to transport across the tropopause. Three of the organic-sulfate profiles shown in Figure 4 appear to be constant with respect to ozone/altitude, three may show a decrease and one shows an increase. The significance of these

slopes is hard to determine without some metric of the measurement uncertainty. From the narrative, it appears that there was a limited (but unspecified) amount of particle composition data at ozone levels above 500 ppbv, so an indication of how significant the measurements > 500 ppbv is necessary to interpret the importance of this slope. Comparing the ATom 2 data in Fig 3 with Fig 1, it appears that there is in fact no significant difference in organic-sulfate particles in the 200nm mode, and actually a significant increase in the amount of organic-sulfate in the 600nm mode. The tropospheric origin of these organic-sulfate particles could also be indicated by their compositional signature, as the authors state on line 143. This is not demonstrated to the reader, even though the data must surely be available.

- It is important to understand that the identification of the tropospheric particles is based on their mass spectra, not the vertical profile. We have changed the text in several places to state this more clearly. We have added a new supplemental figure S2 showing how closely the mass spectra of organic-sulfate particles in the stratosphere match those in the upper troposphere. (We have this in supplemental because mass spectra of the major types of particles in the lower stratosphere have already been published in Murphy et al., 2014.) The text in question has also been revised to point out that ATom3, which has increasing concentrations with ozone, is a special case because of the pyrocumulus injection. The other cases show decreasing concentrations whenever the concentrations are significant. The new table S1 shows that there are many mass spectra of particles in air with more than 500 ppbv ozone for the deployments when the DC8 sampled such air.

Throughout the paper there is no quantitative discussion of the uncertainty or statistics associated with this highly averaged data (in the narrative or in any of the figures). This issue is addressed anecdotally when convenient. For example, in the caption in figure 1 the authors do state that there 'may be artifacts' due to Mie resonance for certain features but other features are robust without any further explanation or justification. If there were other publications describing this data set in detail, some

of these considerations could be addressed through references to these publications, however it appears that these publications are planned for the future.

- We have added a paragraph on uncertainties. The size measurements are described in much more detail in the Brock et al. (2019) reference, which is specifically about the ATom measurements.

Section 5 of the manuscript primarily describes the radiative and chemical impact that arises from the divergent size modes between aerosol of tropospheric and strato-spheric origin. The narrative in section 5 is somewhat disjointed and difficult to follow as it is not always clear how it relates back to the measurements presented in the first sections of the paper. For instance, in section 5.1, the infrared absorption is only discussed in the context of particle volume, yet while the authors acknowledge that particle composition plays a role (line 266) this is not tied back to one of the primary observational findings, that the two size modes have significantly different composition. There are also several results that described in section 5 that are not demonstrated to the reader. For example, on line 329: "The relative contributions to scattering of light at < 240 nm are fairly similar to the surface area panel in Figure 7 except that sizes smaller about 80 nm and larger about 600nm contribute less to UV scattering than they do to surface area." While this statement is quite likely true, it has not been shown and it is hard to quantify what 'fairly similar' describes. A similar statement begins on line 313 "On Figure 7, the sedimentation flux as a function of size would be slightly more skewed to large diameters than the light scattering panel." Such generalized statements need to be shown to be supported by the data.

- The text in section 5.1 has been changed to clarify that while the amount of infrared absorption may change with composition, the size dependence does not. We have added a new figure to the supplemental material with the additional curves about sedimentation and UV scattering. We are trying in the main text not to have figures with too many curves.

[Figure]

Section 6 of the manuscript sets out to describe the relevance of this work to volcanic eruptions or to future geoengineering projects. While the discussion is interesting it is highly speculative and largely unrelated to the data presented in Sections 1-4. For example, in the paragraph starting on line 396 it is implied that differences between the modeled sulfuric acid particle sizes and these measurements (Figure 2) is a consequence of one of the main results that is presumably shown by the data – multiple sources of stratospheric aerosol. However, no details on the initialization for the model run or analysis of what is driving these discrepancies is provided, as a result the comparison in Figure 2 is largely anecdotal.

- The model is described in more detail in the Yu et al. reference and we have added a sentence saying that the model does not distinguish meteoric-sulfuric particles. Finally, the title of the paper "Radiative and chemical implications of the size and composition of aerosol particles in the existing or modified global stratosphere" is misleading. While the aerosol composition data is presented in a limited way, the implications (described in sections 5 and 6) are based entirely on the size distribution of the particles not their composition, even when composition would certainly be important to these implications (e.g. 5.2 Scattering and 5.3 chemistry).

- We feel the title is appropriate. The paper does discuss both size and composition. Although the composition does not enter directly into the radiative properties, it is crucial to the attribution of those properties. Without information about the composition we could not say, for example, what percentage of extinction is due to particles formed in the stratosphere and what percentage due to particles formed in the troposphere. We have added two sentences about this near the beginning of section 6.

Technical Comments: Line 54: "The moles of oxygen were approximately four times sulfur plus about 0.2 times carbon" - rephrase to make it clear where the brackets are $O2 = 4 *(S + 0.2C)$?

- This has been rephrased and a comma added to make it more clear where the brackets belong.

Paragraph starting line 95: It is unclear why the OPC data is needed if PALMs provides sizing?

- More explanation has been added.

Line 174: Using ozone as a proxy for altitude is understandable, but some indication of what an equivalent tropopause relative altitude or range of altitudes that a given ozone mixing ratio corresponds with would be useful.

- This is shown in Figure S1.

Figure 6: What is the unit of measure for 'Approximate net cooling' and how was this calculated?

- The scaling is consistent with the scattering curves. For example, if the "net cooling" curve is 80% that of the light scattered to outer space curve, this means that the cooling of the Earth is 80% of what it would be if only light scattering were considered. Text has been added to section 6.2 stating this. Section 6.2 also describes the calculation.

Figure 7: The units of these plots are unclear. Either provide units, or normalize the data to make it clear that it is relative surface/scatter/volume - Units have been added to the figure.

Figure 8: This figure is confusing. What is meant by the IR-heating trace? Is the peak at 0.5 the most relative cooling per IR heating, and would that mean the least or most IR heating?

- More explanation has been added to the caption. The reviewer is correct, the peak at 0.5 means the most cooling per IR heating.

Figure S1: It would be useful to start the color bar at 250ppbv so that only the points used in this study are highlighted, with a clear distinction for where the particles shown in Figure 3 were found.

- We have changed the color scale on Figure S1 that starts at 250 ppbv. Adding a separate color and symbol size for the 100-250 ppbv range of ozone allows readers to see near-tropopause locations. We've also chosen a color scale that doesn't rely on green-red distinctions.

Anonymous Referee #2

The LMS is an important part of the stratosphere and difficult to observe. Its composition differs from the rest of the stratosphere in many ways due to the mixing of tropospheric and stratospheric air. The study by Murphy et al. is based on in-situ data from aircraft of aerosol particle size distributions and composition of individual particles. Particles were classified depending on their composition to study the history of the particles. Radiative impact was discussed in relation to volcanic eruptions and climate engineering. I find large issues with the authors' interpretation of the data. One reason is the generalization in sections 5-7 based on few data (a single season in a single year). Another is the data on organics, and the claim that the organic aerosol comes from the troposphere only, which contradicts previous work. No uncertainties are presented for the observations as far as I can tell, and there are no statistical analysis to support the claims of trends.

- The reviewer is incorrect that sections 5 to 7 are based on a single season or year. The calculations are not dependent on the season but represent the general consequences of most tropospheric particles being smaller than most stratospheric particles. We picked one season to serve as an example rather than have multiple panels for every figure. Summary data for the other seasons and hemispheres are shown in Figure S4.

I find several shortcomings in the manuscript and cannot recommend publication without changes in the data analysis / interpretation of the data. Major comments

1. Uncertainties and quantification a. How large are the uncertainties in the observations? I don't find any numbers on that. b. How well can one quantify different aerosol

constituents from the PALMS data? c. Is it possible to tell whether there is a trend, as in Figure 4, if there are no statistical analysis to support it?

- We have added a paragraph on uncertainties.

2. There are too little organics in the data compared to previous work, and this is not discussed in relation to those studies. a. There is no discussion on organics after Calbuco. Several other groups have found large amounts of organics in volcanic aerosol. Two examples are Schmale 2010 and Andersson 2013 reported much higher organics (or carbon) to sulfate ratios in volcanic particles. Those data were from AMS and Ion beam analysis. b. Vertical gradient in organics is different from Martinsson 2019, who reported higher carbon abundance deeper into the LMS. I cite from their abstract: ". . .the carbonaceous and sulfurous components of the aerosol in the lowermost stratosphere (LMS) show strong increases in concentration connected with springtime subsidence from overlying stratospheric layers. The LMS concentrations significantly exceed those in the troposphere, thus clearly indicating a stratospheric production of not only the well-established sulfurous aerosol, but also a considerable but less understood carbonaceous component. . ." c. There is no real discussion on wildfire smoke. The smoke from the Aug 2017 fires in western North America is evident in Figure 4. The relatively small impact on the organics after the wildfire in that figure is strange given that the event was almost volcanic sized (Peterson 2018). In Figure 1, it is apparently less organics 1-2 months after the fire than in spring the next year. Why is that? The discrepancy between the data in the manuscript with data from other studies is not discussed as far as I can tell. Identifying particles containing organics is not the same as measuring the mass (of organics or carbon), which other techniques do.

- There are few measurements of organic material in the stratosphere other than from PALMS. Our data seem consistent with the CARIBIC bulk analyses of Nguyen and Martinsson (2007).

- The literature consensus is that volcanic aerosol (excluding ash similar particles immediately after an eruption) is sulfuric acid with little organic content. We have added two references. The papers cited by the reviewer did not definitively find higher organics or carbon in volcanic particles. Schmale et al. 2010 state that their data are only tentative: "Overall, the quantification of organic material in the volcanic plume is subject to uncertainties so that we cannot state whether there is a true increase in organics". In addition, both Schmale et al. and Andersson et al., 2013 ascribe tropospheric particles as a likely cause of organic aerosol in a volcanic plume. This is more consistent with our analysis than the reviewer suggests. To quote Schmale et al. "It is unclear whether the apparent increase in carbonaceous mass might reflect injection of volcanic species or injection of tropospheric species which experienced entrainment into the eruption column." And from Andersson et al "we hypothesize that organic material in entrained air constitutes a significant fraction of the particulate carbon observed in volcanic clouds." If entrainment does occur, the PALMS single particle analysis would identify tropospheric particles entrained into a volcanic plume as tropospheric rather than volcanic.

- The inferences in Martinsson et al. 2019 are based on bulk analysis without size resolution. We think it is clear from this manuscript that at least size-resolved data, if not single particle data, are crucial to understanding particle sources in the lower stratosphere. Previous PALMS data support this, see Murphy et al. (2007). It is important to distinguish the organic content of the overall aerosol, which decreases above the tropopause, with the organic content when considering only the tropospheric particles, which has little vertical gradient. This is shown in the figures in Murphy et al. (2007) and also Figure 6 (was Figure 5) in the current manuscript.

3. Large part of the manuscript is focused on a single season in the Northern Hemisphere (Atom2). It is unclear to me why this is the case. Sulfate concentrations in the LMS varies with season due to both seasonal variation in subsidence from the stratospheric overworld and varying cross TP transport. Stratospheric influence is large in winter and spring, and low in summer and fall. Thus: a. Data presented for a specific

season is representative for that season only, and not for the entire year. b. General climatic conclusions cannot be drawn from a single season in a single year.

- The reviewer is incorrect that the analysis is based on a single season or year. The calculations are not dependent on the season but represent the general consequences of most tropospheric particles being smaller than most stratospheric particles. We picked one season to serve as an example rather than have multiple panels for every figure. Summary data for the other seasons and hemispheres are shown in Figure S4.

4. Data after the Calbuco eruption are almost not discussed at all. The authors mention the eruption and that it had some impact on the sulfuric acid but no more details. I understand that the authors focused on the Northern Hemisphere, but this omission is strange to me. I expect some discussion on data that are included in a manuscript.

- We have added additional discussion of the Calbuco data, especially implications for remote retrievals.

Other comments

1. I would like to see a more clear discussion on the history of sulfuric particles. Meteoric ones from the upper stratosphere, and pure sulfuric from the lowest stratosphere (directly from the tropical lower stratosphere in the BDC).

- There is more discussion of this in a previous paper (Murphy et al., 2014) and we hope to continue to acquire data at higher altitudes that will help further study of the history of the types of sulfuric acid particles.

2. The ExTL has very different composition than the rest of the LMS and (stratosphere). The author never mention the ExTL. Why is that?

- The ExTL concept turns out not to be very helpful in presenting these particular data.

3. I think that the phrase biomass burning shall be changed to wildfire smoke since it comes from PyroCb intrusion(s) to the stratosphere. Biomass burning leads the reader

to believe that it is a general upwelling from diffuse fires instead of PyroCb formation in enormous fires.

- The use of "biomass burning" to refer to smoke particle in the remote atmosphere follows common usage in most of the literature. We have added the modifier "wildfire" where appropriate.

4. I think that the manuscript should have a concise conclusions section after the discussions section.

- For this manuscript we found that the text flowed better if the discussion and conclusions were combined. We have added a short paragraph that summarizes some of the implications for satellite retrievals.

L45. "...The local tropopause and slightly above the altitude of the tropical tropopause. . .". This is not true. The tropical TP is located at âĹij17 km, and the LMS ex- tends to âĹij14-15 km in the extratropics. L135. ". . .The primary source of sulfuric acid in the stratosphere, oxidation of carbonyl sulfide, is similar in the two hemispheres. . .". Do you mean the background stratospheric aerosol? It is still an open question whether SO2 or COS is the most important for the background stratospheric aerosol. L183. Change to: . . .Northern hemisphere than in the Southern Hemisphere. . . L191. I do not understand the statement: ". . .the organic-sulfate particles contain on average about 40 to 80 organic material by mass. . ." L260. ". . .Alumina with the size distribution from rocket emissions was calculated to cause net warming (Ross and Sheaffer, 2014). . ." This is a strange comment in the end of a section on volcanic particles and IR. Are there a sentence missing? L361. Change "will it" to "it will" L469. ". . .the modeled particles have about 65% of the climate impact per unit mass as the observations, 160% of the surface area, and sediment about 60% as fast..." Is this referenced to findings in the manuscript? Are there actual observations of these quantities or do you mean estimated from the PALMS observations?

- Minor text changes suggested here have been made. The definition of the lowermost

stratosphere at follows the references listed. There is actually a lot of spread in the definition of "lowermost stratosphere" by different authors. The question of SO2 or COS as the most important source of sulfuric acid in the background stratosphere has been largely settled (Kremser et al. 2016 and Rollins et al. 2017 references). To quote the review paper by Kremser et al. "OCS makes the largest contribution to the aerosol layer" [italics in original]. The statement about organic-sulfate particles was missing "percent". The text has been modified to better tie the Ross and Sheaffer statement to the text around it. The text has been changed to say that the 65% etc. numbers are compared to those calculated from PALMS observations.

- Additional changes not requested by the reviewers:

- Fine-tuning some data processing has resulted in insignificant changes in the figures and tables. For example, the volumes of meteoric-sulfuric particles and sulfuric particles in the top row of Table 1 have changed from 0.045 and 0.109 to 0.046 and 0.103 respectively. Figure 5 has been revised by averaging over narrower ranges of ozone near the bottom of the profiles where there are many mass spectra and wider range of ozone near the tops of the profile where there are fewer mass spectra. This improves the statistical weighting but the patterns of all of the curves are very similar to the previous version. The revised manuscript also corrects an error in the Figure 5 label which should have been volume instead of mass.

---

## Author Response (AR2)

Reply to reviewer:

The reviewer of the revised manuscript had three comments. The first asked us to "give numbers of uncertainties on size estimations". These are in section 2: "With sufficient averaging (minutes), the volume derived from optical size distributions has an uncertainty propagated from size and flow uncertainties of about +13/-28% in the accumulation mode and up to +/- 50% above 1 µm (Kupc et al., 2018; Brock et al., 2019). Excellent agreement between extinctions calculated from the size distributions and independent extinction measurements indicates that systematic errors may actually be less than this (Brock et al., 2019)." Volume is the most important property of the size distributions for this work. Readers interested in uncertainties for other properties of the size distribution may consult these two references. This comment also asked if the trends in Figure 5 are statistically significant. It is not clear which trends the reviewer was referring to. The increase with altitude in meteoric particles is definitely significant. We have added a sentence saying that more measurements would be needed to determine if the observed seasonal differences are persistent between years. The issue goes beyond just measurement uncertainty in the measurements to how representative any single mission can be. For example, if some concentration in August 2016 was more than in February 2017 is it because concentrations are larger in August or because 2016 had high concentrations?

The second comment asked about mis-classified particles, especially organic-sulfate particles misclassified as meteoric or sulfuric after the pyroCb event. In response to this comment we manually reviewed the classification of a sampling of particles after that event and did not find any erroneous classifications. The meteoric-sulfuric particles in the ATom3 Northern Hemisphere (after the pyroCb) were indeed slightly larger than during other deployments. This is not the result of misclassification. We do not yet understand the reason for the slightly larger size. We have added a paragraph at the end of section 3 illustrating some of the broader features of the composition-resolved size distribution that we are confident in as well as some narrow features in which we are not confident. The reviewer also asked why subsidence shouldn't lead to more meteoric particles during winter and spring than summer and fall. One should remember that we are defining our penetration into the stratosphere with ozone. Subsidence brings down both more meteoric particles and more ozone, so a scatter plot like Figure 5 will not to first order show the effect of subsidence.

The third comment from this reviewer asked why there is less carbon in the meteoric-sulfuric particles in the SH than in the NH. This is discussed in section 4, near line 265.

---

## Author Response (AR3)

Dear Dr. Daniel Murphy,

According to the referee comments and also in my point of view, your manuscript still needs minor revision before acceptance for publication in ACP. I am concerned with the content of 'Section 6 Relevance to volcanic or intentional aerosol injection', which seems not to have a strong linkage with (or supported by) measurement results presented in previous sections. I would suggest that Section 6 be skipped over or reduced and then merged into Section 7. Below are specific comments, some of which are technic.

*Section 6 has been reduced, with less emphasis on intentionally introduced material. Some text has been deleted and some moved into sections 5 and 7.*

L24: The expression 'the surfaces used' is not so clear. Do you mean 'the substances used'?

*Done*

L31: over 60 years since 1961?

*Done.*

L59-60: Do you mean most "single" particles larger than about 110 nm? Are these particles internally mixed, e.g., containing an organic core coated by sulfate or vice versa?

*The text has been modified.*

L89-97: While the UHSA and LAS recorded the data at 1s intervals, how about the sampling frequency of PALMS for particle composition?

*PALMS records single particles, so each particle is associated with a particular time. The averaging needed for PALMS data is discussed in the paragraph starting on line 115.*

L138-139: What radiative transfer model was used? Better to provide some information here, instead only referring to Murphy (2009).

*There is no radiative transfer model required, only Mie scattering. The text has been changed.*

L208-209: It is difficult to follow the discussion here. How are the tropospheric mode and stratospheric particles defined? Please describe the method or threshold used to distinguish the modes (aerosols) between the stratosphere and troposphere here and/or in the caption of Figure 2. If applicable, the same legends can be used in Figure 2 as in Figure 1.

*Text has been added to the Figure 2 caption to better compare to Figure 1. Text has also been added near line 110.*

L244: The phrase 'were both local springtime' can be changed to 'were observed both in local springtime'.

*Done.*

L255: Since ammonium sulfate aerosol has very low volatility, it might not be so meaningful to discuss the equilibrium of ammonia with it. Did you find ammonium nitrate by the PALMS? Otherwise, I would suggest that ammonia be omitted.

*Even very small (~ 1 pptv) amounts of ammonia would be very important for new particle formation, so it is important to constrain the concentration. This is now mentioned in the text/*

L268-269: Does the CESA/CARMA model include these species and associated gas-particle processes? If applicable, what specific species used in the model?

*The CESA/CARM model does include secondary organic chemistry and gas-particle processes (Yu et al., 2015). This is now mentioned in the text/*

L374: Change 'than in the stratosphere than' to 'in the stratosphere than'.

*Done.*

L486-509: It is difficult to follow the discussions in these two paragraphs. While it is declared at the beginning (L488) that infrared absorption is almost independent of size, the infrared effect (L490), relative infrared effect (L492) and infrared absorption (L496) are said to be size dependent. I could not see the infrared effect on net cooling as a function of size from Figure 7 as stated in L489-490, neither high sedimentation rate (downward triangles) in Figure 9. The 'IR heating' appearing in the legend of Figure 9 is unclear.

*The text has been modified here and in the captions of Figures 7 and 9 to clarify the which effects are per unit mass and which are per amount of cooling. The infrared absorption per unit mass is almost independent of size; the infrared absorption per amount of cooling of the Earth is strongly size dependent.*

L547-549: This paragraph seems to be repeating the discussion given in previous section.

*Yes is it, this is a summary section that repeats some key points. The reorganization of sections 6 and 7 may help this.*

Figures 7 and 9: The model and input parameters used to derive these two plots are not well described. What data from the ATom mission are used for the calculation?

*These are model calculations. There are no ATom data in Figures 7 or 9. They are, however, crucial to the discussion of ATom data. The middle panel of Figure 8 is essentially the ATom data in Figure 2 (solid lines in Figure 2) multiplied by the calculations in Figure 7. Figure 9 generalizes some implications of Figure 8 to arbitrary sizes.*

Sincerely,
Jianzhong Ma